# PIWI silencing mechanism involving the retrotransposon *nimbus* orchestrates resistance to infection with *Schistosoma mansoni* in the snail vector, *Biomphalaria glabrata*

**Michael Smith[1], Swara Yadav[2], Olayemi G. Fagunloye[2], Nana Adjoa Pels[2], Daniel A. Horton[3], Nashwah Alsultan[2], Andrea Borns[2], Carolyn Cousin[2], Freddie Dixon[2], Victoria H. Mann[4], Clarence Lee[2], Paul J. Brindley[4], Najib M. El-Sayed[5], Joanna M. Bridger[3], Matty Knight[1,4]\***

1 Howard University, Washington, District of Columbia, United States of America, 2 Division of Science & Mathematics, University of the District of Columbia, Washington, District of Columbia, United States of America, 3 Centre for Genome Engineering and Maintenance, Division of Biosciences, Department of Life Sciences, College of Health, Medicine and Life Sciences, Brunel University, London, United Kingdom, 4 Department of Microbiology, Immunology & Tropical Medicine, Research Center for Neglected Diseases of Poverty, School of Medicine & Health Sciences, The George Washington University, Washington, District of Columbia, United States of America, 5 Department of Cell Biology and Molecular Genetics and Center for Bioinformatics and Computational Biology, University of Maryland, College Park, Maryland, United States of America

\* mathilde.knight@udc.edu, matty_knight@gwu.edu

## Abstract

### Background

Schistosomiasis remains widespread in many regions despite efforts at its elimination. By examining changes in the transcriptome at the host-pathogen interface in the snail *Biomphalaria glabrata* and the blood fluke *Schistosoma mansoni*, we previously demonstrated that an early stress response in juvenile snails, manifested by induction of heat shock protein 70 (Hsp 70) and Hsp 90 and of the reverse transcriptase (RT) domain of the *B. glabrata* non-LTR-retrotransposon, *nimbus*, were critical for *B. glabrata* susceptibility to *S. mansoni*. Subsequently, juvenile *B. glabrata* BS-90 snails, resistant to *S. mansoni* at 25˚C become susceptible by the F2 generation when maintained at 32˚C, indicating an epigenetic response.

### Methodology/Principal findings

To better understand this plasticity in susceptibility of the BS-90 snail, mRNA sequences were examined from *S. mansoni* exposed juvenile BS-90 snails cultured either at 25˚C (non-permissive temperature) or 32˚C (permissive). Comparative analysis of transcriptomes from snails cultured at the non-permissive and permissive temperatures revealed that whereas stress related transcripts dominated the transcriptome of susceptible BS-90 juvenile snails at 32˚C, transcripts encoding proteins with a role in epigenetics, such as PIWI (*BgPiwi*), chromobox protein homolog 1 (*BgCBx1*), histone acetyltransferase (*BgHAT*), histone

**Data Availability Statement:** All relevant data are within the manuscript and its Supporting Information files.

**Funding:** National Science Foundation (US)- http://www.nsf.org (Grant number Award No. 162281) supplied of salary/resources to FD. CBT Knight foundation (US) http://www.cbtknightcancerfoundation.org/ supplied of salary/resources to DAH and MS. DAH was supported by College of Health, Medicine and Life Sciences at Brunel University (UK) www.brunel.ac.uk. The funders had no role in study design, data collection and analysis, decision to publish, or preparation of the manuscript.

**Competing interests:** The authors have declared that no competing interests exist.

deacetylase (*BgHDAC)* and metallotransferase (*BgMT*) were highly expressed in those cultured at 25˚C. To identify robust candidate transcripts that will underscore the anti-schistosome phenotype in *B. glabrata*, further validation of the differential expression of the above transcripts was performed by using the resistant BS-90 (25˚C) and the BBO2 susceptible snail stock whose genome has now been sequenced and represents an invaluable resource for molecular studies in *B. glabrata*. A role for *BgPiwi* in *B. glabrata* susceptibility to *S. mansoni*, was further examined by using siRNA corresponding to the *BgPiwi* encoding transcript to suppress expression of *BgPiwi*, rendering the resistant BS-90 juvenile snail susceptible to infection at 25˚C. Given transposon silencing activity of PIWI as a facet of its role as guardian of the integrity of the genome, we examined the expression of the *nimbus* RT encoding transcript at 120 min after infection of resistant BS90 *piwi*-siRNA treated snails. We observed that *nimbus* RT was upregulated, indicating that modulation of the transcription of the *nimbus* RT was associated with susceptibility to *S. mansoni* in *BgPiwi*-siRNA treated BS-90 snails. Furthermore, treatment of susceptible BBO2 snails with the RT inhibitor lamivudine, before exposure to *S. mansoni*, blocked *S. mansoni* infection concurrent with downregulation of the *nimbus* RT transcript and upregulation of the *BgPiwi* encoding transcript in the lamivudine-treated, schistosome-exposed susceptible snails.

## Conclusions and significance

These findings support a role for the interplay of *BgPiwi* and *nimbus* in the epigenetic modulation of plasticity of resistance/susceptibility in the snail-schistosome relationship.

## Author summary

Progress is being made to eliminate schistosomiasis, a tropical disease that remains endemic in the tropics and neotropics. In 2020, WHO proposed controlling the snail population as part of a strategy toward reducing schistosomiasis, a vector borne disease, by 2025. The life cycle of the causative parasite is, however, complex and in the absence of vaccines, new drugs, and access to clean water and sanitation, reduction of schistosomiasis will remain elusive. To break the parasite's life cycle during the snail stage of its development, a better understanding of the molecular basis of how schistosomes survive, or not, in the snail is required. By examining changes in the transcriptome at the host-pathogen interface in the snail *Biomphalaria glabrata* and *Schistosoma mansoni*, we showed that early stress response, manifested by the induction of Heat Shock Proteins (Hsps) and the RT domain of the non-LTR retrotransposon, *nimbus*, were critical for snail susceptibility. Subsequently, juvenile *B. glabrata* BS-90 snails, resistant to *S. mansoni* at 25˚C were observed to become susceptible by the F2 generation when maintained at 32˚C, indicating an epigenetic response. This study confirms these earlier results and shows an interplay between PIWI and *nimbus* in the anti-schistosome response in the snail host.

## Introduction

The freshwater snail, *Biomphalaria glabrata*, is an obligate intermediate host of the trematode, *Schistosoma mansoni*, the causative agent of the neglected tropical disease (NTD) schistosomiasis in neotropical regions. At least 600 million people, mainly in sub-Saharan Africa, are at

risk for schistosomiasis, a number that remains excessively high, despite efforts to control transmission of the disease [1]. This disease causes widespread chronic morbidity and male and female infertility. Specifically, infections caused by the species, *Schistosoma haematobium* may result in bladder cancer and female genital schistosomiasis. The latter exacerbates transmission of sexually transmitted diseases including HIV (AIDS) [2].

The disease burden from schistosomiasis is probably underestimated, and it has been suggested that the number of infected individuals exceeds 400 million [3]. On the other hand, there are other estimations that claim 230 million people worldwide are infected with *S. mansoni* [1]. These numbers are often underestimated due to the inability of current diagnostic methods to detect light infections [4]. There are few defenses against schistosomiasis, mainly because the residents of infected areas lack sufficient infrastructure to properly combat this disease [3]. An integrated control approach, implementing mass chemotherapy and molluscicides has made a difference in breaking the complex life cycle of the parasite but without new effective drugs and vaccines to prevent re-infection in treated areas, the cycle of repeated infection is the norm, thereby making the longterm control of schistosomiasis elusive [5,6]. Because of recent projections made by the World Health Organization to eliminate schistosomiasis by 2025 and coupled with recent concerns of the spread of the disease into Mediterranean countries of Europe, alternative approaches focusing mainly on blocking the development of the parasite in the snail host are aggressively sought [7–9].

Molecular mechanism(s) involved in shaping the relationship between the parasite and its obligate intermediate snail host, *Biomphalaria glabrata*, remains largely unknown. However, information that will assist in clarifying some of the mechanisms of host/parasite interactions is steadily being amassed. Reference genome sequences for all three organisms (human, schistosome and snail) pertinent to transmission of schistosomiasis are now available [10–12]. Additionally, genes that underlie resistance and susceptibility phenotypes in *B. glabrata* to *S. mansoni* infection are being identified [13–15], as are transcripts encoding larval (miracidia) parasite proteins that are expressed at the snail/parasite interphase [16,17]. Both snail and parasite determinants are involved in a complex and dynamic innate defence system that either rejects or sustains the successful development of the intra-molluscan stages of the parasite [13,15].

Previously, we demonstrated that juvenile *B. glabrata*, that are either resistant or susceptible to *S. mansoni*, display a differential stress response after early exposure to wild type but not to irradiated *S. mansoni* miracidia. The stress response observed in the susceptible juvenile snail was manifested by the early induction of transcripts encoding heat shock proteins (Hsp) 70, Hsp90 and the reverse transcriptase (RT) domain of the *B. glabrata* non-LTR-retrotransposon, *nimbus* [18]. Furthermore, the non-random relocalization of the Hsp70 gene loci in interphase nuclei preceded transcription of the corresponding Hsp70 transcript in the susceptible but not in the resistant snail, indicating that in-coming schistosomes possess the ability to orchestrate in a rapid and systemic fashion, the genome remodeling of juvenile susceptible snails soon after infection [19]. We also demonstrated that resistance in the juvenile BS-90 snail stock was a temperature dependent trait. Thus, when cultured at room temperature (25˚C), juvenile BS-90 snails remained consistently resistant to infection. However, when cultured at 32˚C for several generations ($F_I$ to $F_3$), the progeny juvenile snails were phenotypically susceptible [20], indicating a plastic epigenetic control over resistance.

Other studies that have used adult (>7 mm in diameter) snails instead of juveniles have suggested that ability to alter the resistance of the BS-90 to infection at elevated temperature might be a strain-specific trait [21]. In recent studies, resistant BS-90 snails were found to be susceptible when exposed to *S. mansoni* as neonates [22]. In general, adult snails are less susceptible to infection [23]. Furthermore, in some stocks of *B. glabrata*, for example the 93375

strain, the juveniles are susceptible but become resistant to the same strain of *S. mansoni* as young adults (at the onset of fecundity), and once egg laying ceases and amoebocyte accumulations disappear in the pericardial wall, revert to the susceptible phenotype [23,24].

To further investigate the molecular basis of susceptibility plasticity, notably in the BS-90 snail, we have undertaken a comparative analysis of the transcriptomes of juvenile BS-90 snails cultured for several generations at either permissive (32˚C) or non-permissive (25˚C) temperatures, aiming to obtain candidates for pathway(s) that lead either to resistance or susceptibility. This investigation and the findings are detailed below.

## Materials & methods

### Snails

The BS-90 snail isolated from Salvador, Brazil, is a wild-type pigmented snail that is resistant to both Old and New World strains of *S. mansoni* (NMRI strain) either as a juvenile or as an adult snail at 25˚C [25,26]. The NMRI snail is an albino susceptible snail that was derived from a cross between the wildtype Puerto Rican snail and a highly susceptible Brazilian albino snail [27]. The BB02 snail is a susceptible pigmented wildtype snail from Brazil whose genomic DNA sequence was recently reported [10]. The susceptible snails (NMRI and BBO2) are highly susceptible as juveniles but the degree of susceptibility, especially in the NMRI stock, is variable as an adult snail [28].

### Snail husbandry and *S. mansoni* infections

BS-90 stocks were cultured at 32˚C as described [20]. Exposures of BS-90 snails cultured at 32˚C to miracidia were performed using juvenile progeny, $F_1$- $F_2$, <4 mm in diameter that were bred at the elevated temperature. Briefly, BS-90 snails were cultured either at 25˚C or 32˚C in freshly made artificial pond water ([www.afbr-bri.com](www.afbr-bri.com)) and fed *ad libitum* with either romaine lettuce or snail gel food [29]. Juvenile snails or egg clutches were transferred from 25˚C to 32˚C and maintained in groups of 3 or 4 in fresh water (100 ml) in beakers maintained in a water bath at 32˚C. The temperature inside the water bath was monitored daily to maintain 32˚C for the duration of the experiment. The snails were cleaned weekly making sure that pre-warmed (32˚C) water was used to clean the snails. Detritus including dead snails and decayed lettuce leaves were removed daily. The egg clutches from these snails (produced at 32˚C) were collected and their progeny were maintained at 32˚C until they had grown to 3 to 4 mm in diameter (juvenile snails) before exposure to miracidia at 25˚C. The juvenile BS-90 snails, 3 to 4 mm in diameter, were maintained for two generations at either 32˚C or 25˚C and RNA prepared from 0 and two-hour infected $F_2$ progeny as described [20]. Snails were exposed individually to the 10 to 12 miracidia in wells of a 12-well tissue culture plate (Greiner Bio-One, North Carolina, USA) at room temperature. Miracidia were hatched from eggs recovered from the livers of mice which had been infected with *S. mansoni* (NMRI strain) for seven weeks [30].

Exposed snails (not used for RNA) were maintained at 25˚C and examined for cercarial shedding from four to 10 weeks later. Susceptible BBO2 and NMRI *B. glabrata* snails utilized in this study were exposed as juveniles (described above) to freshly hatched miracidia. To determine patency (cercarial shedding), individual snails were immersed in one ml nuclease-free water in 12-well plates and directly exposed to a light source for 30 to 60 minutes at room temperature, after which snails were removed from the wells. Cercariae released from individual snails were counted after adding a few drops of Lugol's iodine solution to each well. After shedding, snails that were patent were euthanized by immersion in 95% ethanol; non-shedding snails were incubated for up to 10 weeks at 25˚C and checked weekly for patency.

## RNA sequencing, assembly, and annotation

Total RNA was isolated by RNAzol (Molecular Research Center, Inc. Cincinnati, OH) from resistant (25˚C) and susceptible (32˚C) BS-90 snails. BS-90 snail transcriptomes were generated from polyA$^+$ RNA isolated from pooled intact 2 hr exposed juvenile snails maintained either at the non-permissive temperature (25˚C), or at the permissive temperature (32˚C), on an Illumina HiSeq 100. Following RNAseq Illumina sequencing, *de novo* assembly of the transcriptome was performed using Trinity. Functional annotation of the contigs was performed with Trinotate that included Gene Ontology assignments when possible, Pfam domain identification, transmembrane region predictions (TmHMM), and signal peptide predictions (signalP). Differential expression (DE) analyses were performed by using DESeq and identified differentially expressed contigs between BS-90 snails from 32˚C and 25˚C. The analyses of these contigs revealed the presence of known specific genes coding for several stress related and other transcripts (Table 1).

DE analysis was carried out using DEseq on Trinity assembled de novo transcripts and > than 2000 genes were identified. Representative up and down regulated transcripts in resistant (25˚C) versus susceptible (32˚C) juvenile BS-90 snails were annotated subsequently by using the assembled *B. glabrata* BBO2 susceptible stocks sequenced genome in Vector Base and NCBI as reference [10]. To select robust candidate transcripts that underscore either *S. mansoni* resistance or susceptibility phenotypes, wild type susceptible reference BBO2 and resistant snail BS-90 stocks were used to perform real time qPCR to validate differential expression of selected transcripts. A standard BLASTp was performed to further validate the annotation of the 897 amino acid sequence of *BgPiwi* (Accession No XP_013089248.1) followed by a SMART BLAST to explore the phylogeny of the *B. glabrata* homolog to other PIWI encoding transcripts in the public domain. The complete RNA-seq data sets showing the transcriptome profiles of changes observed in schistosome juvenile BS90 snails at either permissive (32˚C) or non-permissive (25˚C) temperatures are provided as FASTQ files in the SRA database within NCBI with Bioproject ID PRJNA687288.

## Phylogenetic tree construction

The *BgPiwi* homologs identified using SMART Basic Local Alignment Search Tool (BLAST) were aligned using CLUSTAL with a BLOSUM matrix without any manual editing of the multiple sequence alignment (MSA). The MSA was passed to a maximum-likelihood search with phyloML using a SPR tree search from a Neighbour-Joining starter and a 100-replicate bootstrap.

## Two step qPCR

Differential expression of the selected transcripts identified from the single pass RNA-seq dataset generated from resistant (25˚C) and susceptible (32˚C) juvenile BS-90 snails were (as mentioned above) further validated in representative resistant BS-90 and susceptible BBO2 snail stocks. Real time qPCR was also performed by the lab maintained susceptible NMRI snail to determine whether transcription patterns, up *versus* down, were reproducible. Experiments were performed using multiple individual snail samples (> 10) for each of the snail stocks, and independently (3 to 5 biological replicates) by different investigators with triplicate samples for each time point. Total RNA was isolated from juvenile snails that were either unexposed (0 hour) or exposed to *S. mansoni* for 30 min, 1, 2, 4 and 16 hours at 25˚C as described [31]. cDNA was prepared from the total RNA after residual contaminating genomic DNA was removed by treatment of the RNA with RNase-free DNase (RQI Promega WI) before performing the qPCR assays [25]. Quantitative PCR was performed with forward and reverse gene

**Table 1. Differential expression of downregulated (green) and upregulated (red) transcripts from RNA-seq data of *B. glabrata* BS-90 snail stocks cultured either at permissive (32˚C) or non-permissive (25˚C) temperature.**

| Assembled transcript ID | Fold change | Gene product/identity |
|---|---|---|
| *Up-regulated* | | |
| comp687935_c0 | 77 | Heat shock 70 kDa protein PPF203 |
| comp680371_c0 | 77 | Heat shock protein homolog ECU03$_{0520}$ |
| comp833304_c0 | 72.8 | Chaperone protein DnaK |
| comp729179_c0 | 51.7 | Heat shock 70 kDa protein 4 |
| comp53294_c0 | 40.9 | Heat shock 70 kDa protein |
| comp116707_c0 | 40.2 | Heat shock protein 83 |
| comp174809_c1 | 21.6 | Alpha−crystallin B chain |
| comp22367_c0 | 12.6 | Heat shock protein 90 homolog |
| comp567369_c0 | 8 | Ferric−chelate reductase 1 |
| comp173740_c0 | 6.4 | Heat shock 70 kDa protein |
| comp396056_c0 | 6.1 | Apoptosis 2 inhibitor |
| comp175527_c0 | 5.3 | Heat shock protein 70 B2 |
| comp248752_c0 | 5.1 | Heat shock protein 68 |
| comp122866_c0 | 5.0 | Heat shock protein beta−1 |
| comp144872_c0 | 5.0 | Alpha−crystallin B chain |
| comp181825_c1 | 4.8 | Histone−lysine N−methyltransferase, H3 lysine−79 specific |
| comp319764_c0 | 3 | Universal stress protein Rv2026c/MT2085 |
| *Down-regulated* | | |
| comp129067_c0 | -2.6 | Heat shock protein Hsp−16.2 |
| comp185349_c3 | -2.8 | Protein arginine N−methyltransferase 1 |
| comp173107_c0 | -2.8 | Monocarboxylate transporter 9 |
| comp43678_c0 | -2.9 | Histone RNA hairpin−binding protein |
| comp142687_c0 | -3.1 | Mitogen−activated protein kinase kinase kinase3 |
| comp100349_c0 | -3.0 | Histone deacetylase 1 |
| comp86764_c0 | -3.1 | Piwi−like protein 1 |
| comp146653_c0 | -3.3 | Cathepsin B |
| comp143747_c0 | -3.9 | Apoptosis regulator BAX |
| comp375043_c0 | -3.5 | Calcium−regulated heat stable protein 1 |
| comp126708_c0 | -3.6 | FAS−associated factor 1 |
| comp169361_c0 | -4.3 | Macrophage mannose receptor 1 |
| comp153455_c0 | -4.4 | Histone deacetylase 2 |
| comp54296_c0 | -4.5 | Chaperone protein ClpB |
| comp43231_c0 | -5.3 | Chromobox protein homolog 1 |
| comp887090_c0 | -6.1 | Tudor domain−containing protein 12 |
| comp182899_c1 | -7.0 | Peroxiredoxin−4 |
| comp66580_c0 | -7.4 | Protein arginine N−methyltransferase 1 |
| comp920619_c0 | -8.1 | Interferon−related developmental regulator 1 |
| comp178355_c0 | -9.6 | Histone acetyltransferase KA T6B |
| comp599_c0 | -9.6 | MAU2 chromatid cohesion factor homolog |

specific primers corresponding to the selected transcripts normalized against the expression of the myoglobin as a reference, as previously described [31]. Oligonucleotide primers (forward and reverse, Table 2) were designed for selected transcripts identified from the annotated sequences of RNA-seq datasets from 2hr *S. mansoni* exposed BS-90 snails cultured at either permissive (32˚C) or non-permissive temperatures (25˚C). The amino acid sequences

**Table 2. Selected transcripts, accession numbers, and corresponding F and R primers utilized for qPCR.**

|    | Description | Sequence | Accession Number |
|----|-------------|----------|------------------|
| *1.* | *BgPiwi* | 5':TTGCAAAATGGGCGGTGAAG | XP_013081375 |
|    |          | 3': TGACGAACTGACTGGCTCAC | |
| 2. | *BgHDAC* | 5':CCACATAAGGCCACAGCAGA | XP_013075422.1 |
|    |          | 3': TAGTACTTGCCCTTGCCTGC | |
| 3. | *BgCBX1h-1* | 5': CAACGTGCATTTAAGGCGGA | XP_013064838.1 |
|    |          | 3': CACTGCTGTCTGTAGCACCT | |
| 4. | *BgHAT* | 5': CGGCGGCATTTATCTTGGTG | XP_013062341.1 |
|    |          | 3': TGTCAATGTGGCGTCGAAGA | |
| 5. | *BgMT* | 5': CATAGTCCGGTTGGTGCAGA | XP_013081375 |
|    |          | 3':GCAGTTGGTAGCAGCAAGAGA | |
| 6. | *nimbusRT* | 5':GCTCCATTAAACCGAACAGAC | EF413179 |
|    |          | 3':CCCCGTAGATCATTGCTAAC | |

corresponding to the coding DNA sequence (CDS) of the following transcripts: PIWI like protein (*BgPiwi*), chromobox protein homolog 1 (*BgCBx1*), methyltransferase (*BgMT*), Histone Acetyl Transferase (*BgHAT*) and histone deacetylase (*BgHDAC*) were utilized to interrogate the reference *B. glabrata* (BBO2 stock) genome in GenBank by using BLAST in NCBI. Amino acid sequences of *B. glabrata* CDS showing significant (E value = $<10^{-4}$ and with >25% amino acid sequence identity) homology to other transcripts were converted to the nucleotide sequences and gene specific primers were designed using primer BLAST in NCBI. Forward (F) and reverse primers (R) for qPCR analysis were obtained from Eurofins Genomics (Louisville, KY) after the exclusion of sequences for *S. mansoni* to avoid any possible amplification of parasite RNA during qPCR analysis.

Two-step qPCR was utilized to quantitatively assess expression of the selected transcripts using 500 ng of cDNA as template. SYBR Green PCR Master Mix kit (Applied Biosystems, Thermo Fisher Scientific, Woolston Warrington, UK), with 15 μM of forward and reverse primers were used to evaluate the temporal expression of PIWI (*BgPiwi*), Chromobox protein homolog 1 (*BgCBx1*), HDAC (*BgHDAC*), HAT (*BgHAT*) and methyl transferase (*BgMT*). Each sample was run in triplicate and reactions normalized against the constitutively expressed myoglobin reference gene in a 7300-thermal cycler (Applied Biosystems). Primers were designed from the CDS of the selected transcripts and optimized for qPCR as described previously [31]. Relative quantitative expression of the genes of interest between resistant and susceptible snails was evaluated by the ΔΔCt method. The resulting fold change in expression of the genes of interest normalized against the signal for myoglobin were calculated by using the formula [32],

$$\text{Fold difference} = 2^{-\Delta\Delta Ct2} = 2^{-[(Ct_{gene,test}-Ct_{myoglobin,test})-(Ct_{gene,control}-Ct_{myoglobin,control})]}$$

The variance of each of the data groups was determined using F-test. Also, Shapiro-Wilk test and Kolmogorov-Smirnov test were used to test for normality and lognormality. Differences between the groups were assessed using Student's *t* test, Welch's *t* test and 2-way analysis of variance (ANOVA) wherever relevant by comparing the differential expression (delta-Ct value) of the transcripts among treatment and control groups. A *p*-value of <0.05 was considered to be statistically significant, with level of significance denoted as follows, ****, $p \leq 0.0001$, ***, $p \leq 0.001$, **, $p \leq 0.01$, *, $p \leq 0.05$, and ns, $p > 0.05$.

## *BgPiwi* transcript silencing by dsRNA and siRNA

To investigate the functional role of *BgPiwi* expression in *B. glabrata* susceptibility to schistosome infection, the transcription of *BgPiwi* was silenced by soaking juvenile BS-90 snails in either dsRNA-, or siRNA- complexed with PEI [33]. Double–stranded (ds)RNA corresponding to *BgPiwi* was synthesized by using an *in vitro* transcription kit with a purified *BgPiwi* PCR product containing T7 sequences (sense and antisense) as template according to the manufacturer's instructions (MEGAscriptT7, ThermoFisher Scientific Inc.) [34]. Off-target silencing of the transcript encoding *BgPiwi* in the resistant BS-90 snail was evaluated by soaking snails in universal mock siRNA- PEI complexes as control (MISSION siRNA Universal negative control#1). Knock-down of the *BgPiwi* transcript in the resistant BS-90 snail (cultured at 25°C) was done as follows; juvenile snails were placed in 1.0 ml nuclease-free $dH_2O$ containing either 300 ng dsRNA: 1.0 µg PEI nanoparticle complexes or 775 ng siRNA: 1.0 µg PEI nanoparticle complexes. The complexes were prepared as follows: in a 1.5 ml capacity microcentrifuge tube, 1 µg of PEI (Sigma Aldrich), branched with average molecular weight 25000, in 500 µl nuclease-free $H_2O$ was added slowly, drop-wise, to two different siRNAs (Sigma Millipore) (start on target sequence CDS position 2393bp- sense: GAACCAUUGUGGAUCAAAU/anti-sense: AUUUGAUCCACAAUGGUUC; (start on target sequence at CDS position 2403bp sense: GGAUCAAAUAAUUACGAA/anti-sense: UUUCGUAAUUAUUUGAUCC) diluted in 500 µl before mixing vigorously for 10 seconds at room temperature. Both duplex siRNAs/PEI nanoparticles corresponding to the *BgPiwi* transcript were utilized simultaneously in a single tube. Samples of siRNA/PEI complexes (total of 1.0 ml in microcentrifuge tubes) were incubated at room temperature for 20 minutes before placing individual juvenile snails in the mixtures. Holes were punched in lids of the closed microcentrifuge tubes containing snails in siRNA/PEI complexes before incubating overnight at room temperature. Control tubes, incubated in parallel, contained the following samples, a) *BgPiwi* siRNAs without PEI, b) PEI only without *BgPiwi* siRNAs and c) Mission Universal mock siRNA/PEI complexes (Sigma Millipore). RNA was isolated as described above from washed transfected snails before utilizing for qPCR as described above. For each assay, quantitative expressions of *BgPiwi* and *nimbus* RT transcripts (normalized against myoglobin expression) were evaluated with and without *S. mansoni* infection by qPCR using forward and reverse primers corresponding to either transcript. Transfected snails (with and without infection) that were not investigated by RNA-based assays were maintained in at 25°C as above and monitored for cercarial shedding at 4, 6, or 10 weeks later. The silencing of *BgPiwi* with dsRNA/PEI was evaluated in three, and with siRNA/PEI, in five biological replicates, respectively.

## Treatment of susceptible BBO2 snails with reverse transcriptase inhibitor, lamivudine

Given that the central role of PIWI involves silencing of endogenous mobile elements, such as *nimbus*, a non-LTR retrotransposon in the genome of *B. glabrata* [10,35], we examined the modulation of expression of the transcript encoding the RT domain of *nimbus* in the following categories of susceptible BBO2 snails—a) normal snails, b) snails treated overnight at room temperature with lamivudine at 100 ng/ml (Sigma Aldrich, St. Louis, MO) and c) BS-90 resistant snails treated with siRNA corresponding to *BgPiwi*/PEI complexes, as described above. Snails in these categories (a to c) were either unexposed (0) or exposed (individually) to 10 miracidia for 120 min at 25°C. Before exposure, individual snails incubated in either lamivudine or *BgPiwi* siRNA/PEI complexes were washed twice with water before transferring to 2.0 ml water in 6-well tissue culture plates to which freshly hatched miracidia (isolated from 7 weeks infected mouse liver homogenate) was added and maintained for 120 min at room

temperature. Exposed and unexposed snails from either lamivudine-treated susceptible BBO2 or BS-90 *siRNA BgPiwi*/PEI- treated snails were either frozen immediately at -80˚C in RNAzol until required for RNA isolation or, if not used for RNA preparation immediately, transferred into 500 ml beakers containing aerated tap water and maintained as described above at room temperature and evaluated for cercarial shedding at week 4, 6 and 10 post-exposure. For comparison, susceptible snails were also either pre-treated as described above, or after two weeks post-parasite exposure, with another RT inhibitor BPPA (Santa Cruz Biotechnology Inc., CA) that specifically inhibits the catalytic RT domain of human telomerase (hTERT).

## Examining genome organization, relocation of the *PIWI* locus, in susceptible and resistant snails following exposure to *S. mansoni* miracidia

Fluorescence *in situ* hybridisation (FISH) was performed using a probe derived from *B. glabrata* Bacterial Artificial Chromosome (BAC) libraries for the *PIWI* locus. The DNA probe was labelled by nick translation (BioNick Invitrogen, UK) as described [36,37] and incubating for 45–50 mins. The probe was precipitated with 1μg of labelled BAC DNA [36], 80 μg of *B. glabrata* genomic DNA and 9 μg of herring sperm DNA. These components were dissolved in 48 μL of hybridisation mix at room temperature overnight, this amount can be used for up to four slides. Preparation and fixation of samples followed the protocol described previously [19]. Snail shells were crushed using a microscope slide and the ovotestes excised using needle-nose forceps. Each ovotestis was placed in a microcentrifuge tube containing 0.05 M KCl, macerated using a tissue grinder (Axygen, UK) and incubated in solution for 30 min at room temperature. Samples were then centrifuged at 200g for 5 min and supernatant discarded. Methanol:acetic acid [3:1, v/v] was added dropwise, with agitation, to the samples. Once 0.5 mL of fixative was added, the samples were incubated at room temperature for 10 min before centrifuging again and discarding the supernatant, this fixation step was repeated twice with the final fix volume being 100 μL. Slides were also prepared by misting the slide with water vapour and then dropping 20 μL of a sample from a height onto the slide and allowing the slide to dry on a slide drier. The slides were aged by placing into a 70˚C oven for 60 min then were taken through a dehydration series of 70%, 90% and 100% ethanol, spending 5 min in each solution. Slides were dried and warmed up to 37˚C on a slide dryer alongside 22x22 coverslips in preparation for probe addition. Probe denaturation was performed at 75˚C for 5 min and then allowed to reanneal for 20 min at 37˚C before use. Hybridisation of samples and probe was performed using the Top Brite automatic slide hybridiser (Resnova, Italy). Eleven μL of probe was applied to a coverslip and the slide with the sample brought to the coverslip and the coverslip sealed to the slide using rubber cement (Weldtite). The Top Brite was set for 2 min at 37˚C, then 2 min at 75˚C, and then 30 min at 37˚C. Once the slides had been returned to 37˚C, they were transferred to a humidified chamber at 37˚C for 72 hours. Post- hybridisation, the rubber cement was removed, and coverslips allowed to detach in the first wash. The washes were performed at 42˚C in 2x SSC three times for 5 min each. A blocking solution, made of 4% BSA (Sigma Aldrich, UK) in 2x SSC, was prepared. After slides were removed from the third wash the excess solution was drained and 100 μL of blocking solution was added and the slides covered with parafilm. Slides were maintained in a humidified chamber at room temperature for 30 min. Streptavidin-Cy3 was diluted 1:200 in 1% BSA in 2x SSC. After the blocking solution was removed, 100 μl of streptavidin-Cy3 solution was placed on each slide, covered in parafilm and incubated in a humidified chamber at 37˚C for 30 min. After the streptavidin-Cy3 incubation the slides were washed sequentially in 2x SSC for 5 min, 1x PBS + 0.1% Tween 20 (Sigma Aldrich) for 1 min, and 1x PBS for 1 min. Lastly, the slides were rinsed in sterile water before counterstaining with 4',6-diamidino-2-phenylindole (DAPI) in mountant (H1200, Vectorshield).

## Image analysis

Images of nuclei were captured either with an Olympus BX41 fluorescence microscope with a greyscale digital camera (Digital Scientific, UK) and the Smart Capture 3 software (Digital Scientific, UK) or a Leica DM4000 using a Leica DFC365 FC camera and the Leica Application Suite (LAS) imaging software. At least 50 nuclei were imaged for each condition and processed via erosion script analysis [38,39] to assess gene loci positioning by using greyscale images and measuring the intensity of DAPI and fluorescence in situ hybridization (FISH) signal, using the DAPI to outline the nuclei to create five shells of equal area so that the intensity of the DAPI and FISH signal can be measured and the FISH signal normalised for position by dividing by the DAPI signal, averaged for the 50 images. Unpaired, equal variance, Student's $t$ tests were performed to ascertain significant differences in gene loci positioning within the different shells.

## Results

### Transcripts encoding PIWI (*BgPiwi*), HDAC (*BgHDAC*), chromobox protein homolog 1(*BgCBx1*), histone acetyl transferase (*BgHAT*) and metallotransferase (*BgMT*) are differentially regulated in resistant BS-90 and susceptible BBO2 snails

To confirm and validate results of differential expression (DE) obtained from the denovo single-pass sequencing of RNA isolated from *S. mansoni* exposed juvenile BS-90 $F_2$ snails, which were cultured either at non-permissive (25˚C), or permissive (32˚C), temperatures, we investigated the expression of selected transcripts that have a role in epigenetics by two-step qPCR performed with RNA isolated from several different individual juvenile *B. glabrata* snails that were either resistant (BS-90 cultured at 25˚C) or susceptible (BBO2 and BS-90 cultured at 32˚C) to *S. mansoni* infection. Since the objective of this study was to identify robust candidate transcripts that underscore either *B. glabrata* resistance or susceptibility to *S. mansoni*, we chose to confirm the DE regulation of the transcripts identified between permissive (32˚C) and non-permissive (25˚C) BS-90 snails from the RNA-seq data set by using wildtype representative resistant (BS-90) and susceptible (BBO2) snails. This choice was also made because a reference genome sequence exists for the BBO2 stock, making it an invaluable resource for all follow up molecular studies on the *B. glabrata* snail. The investigation of temporal (0, 30 min, 1, 2, 4, and 16 hours) expression of *BgPiwi* in either juvenile resistant BS-90 (25˚C) or juvenile susceptible BBO2 following exposure to *S. mansoni* by qPCR showed that the transcript encoding the *B. glabrata* PIWI-like protein (Isoform 1 Accession number XP_013081375) was upregulated between 30 min and 2 hour (2- to 7-fold) post-exposure in the resistant BS-90 but not in the susceptible BBO2 snail (Fig 1). Fig 2 shows the temporal expression of the transcript encoding *BgHDAC* (Accession XP_0130754221.1). Results likewise showed upregulation of this transcript (1.8-fold) in the juvenile resistant BS-90 but not in the juvenile susceptible (BBO2) snails 2 hours after *S. mansoni* exposure. Similarly, as shown in Figs 3–5, results demonstrated that transcripts encoding, the chromobox protein homolog 1 (Fig 3, *BgCBx1*), histone acetyl transferase (Fig 4, *BgHAT*) and metallotransferase (Fig 5, *BgMT*) were also upregulated in BS-90 resistant but not the susceptible BBO2 snails following *S. mansoni* infection. Inductions of up to 13- and 10-fold for transcripts encoding the chromobox protein homolog 1 and MT, respectively (between 2 and 4 hours post exposure) were observed in exposed juvenile resistant BS-90 but not their exposed juvenile BBO2 susceptible counterparts.

A graphical flow chart summarizing the experimental design and results is shown in Fig 6.

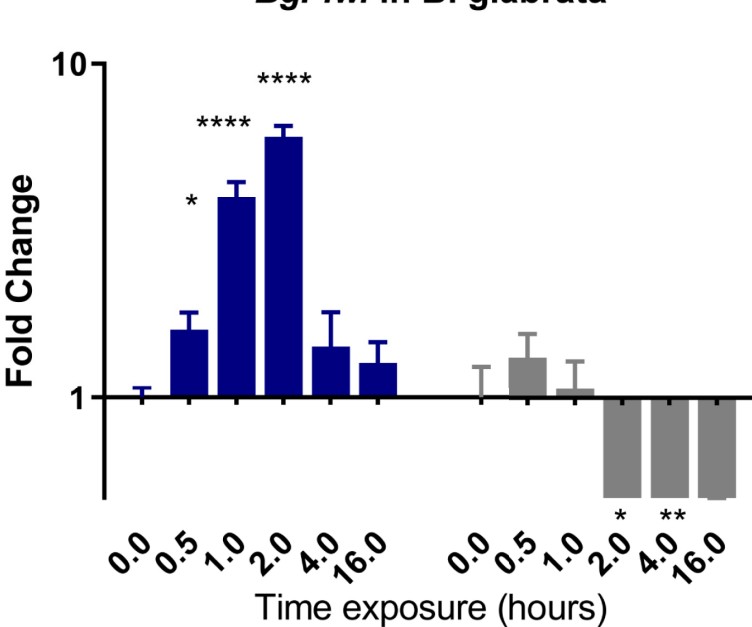

**Fig 1. qPCR analysis of RNA from resistant BS-90 (blue histogram) or susceptible BBO2 (gray histograms) juvenile snails unexposed (0) or exposed for increasing intervals (30 seconds to 16 hours) to *S. mansoni* miracidia.** Histograms show expression of the *BgPiwi* encoding transcript in snails at each time point from five biological replicates. Note the increase in fold change in the resistant BS-90 compared to the susceptible BBO2 snails after parasite infection. Significant expression normalized against expression of the myoglobin encoding transcript was measured by 2-way ANOVA and is indicated by number of asterisks on each histogram where ****, indicates the most significant value $p \leq 0.0001$, *** $p \leq 0.001$, ** $p \leq 0.01$, * $p \leq 0.05$, ns $p > 0.05$.

## The protein sequence of *BgPiwi* is highly conserved

A standard BLASTp was performed to further validate the annotation of the 897 amino acid sequence of *BgPiwi* (Accession No XP_013089248.1) that was identified from the RNA-seq data set. This was followed by a SMART BLAST to explore the phylogeny of the *B. glabrata* homolog to other PIWI encoding transcripts in the public databases. Results revealed that the *BgPiwi* encoding transcript represents a single copy gene in the *B. glabrata* genome that is translated into two protein isoforms, 1and 2. All data presented herein deals only with the differential regulation of the transcript encoding protein isoform1 and not 2. Results of multiple sequence alignment of the *BgPiwi* isoform1 amino acid sequence to those from other organisms showed that the snail protein is highly conserved with significant matches to the vertebrate (human, mouse, zebra fish) orthologs compared to invertebrate PIWI proteins in *Drosophila melanogaster* and *Caenorhabditis elegans* (Table 3).

Indeed, a phylogenetic tree conducted by the neighbor -joining method, confirmed that the *B. glabrata* protein was more closely related to *C. elegans* than to *D. melanogaster* (S1 Fig).

## siRNA corresponding to *BgPiwi* knocks down expression of the PIWI encoding transcript rendering resistant BS-90 snails susceptible

Since the findings revealed that the transcript encoding *BgPiwi* was upregulated in juvenile resistant BS-90 snails after *S. mansoni* infection, we proceeded to examine the modulation of

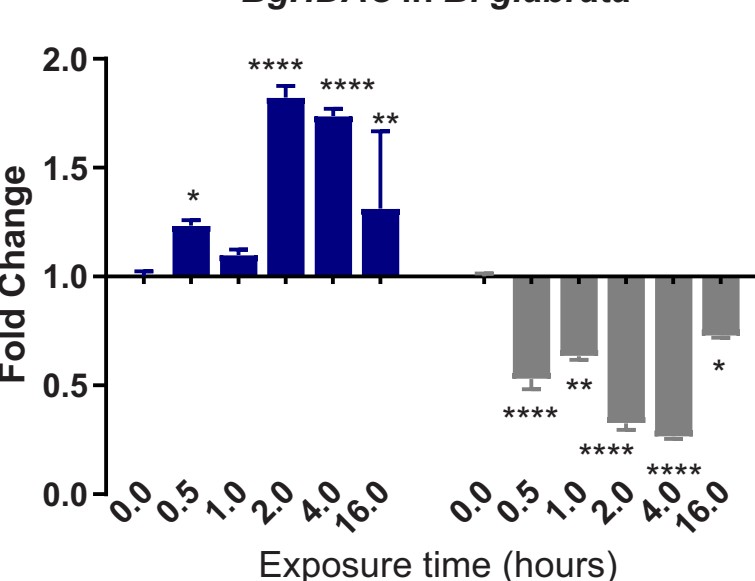

**Fig 2. qPCR analysis of RNA from either resistant BS-90 (blue histogram) or susceptible BBO2 (gray histograms) juvenile snails unexposed (0) or exposed for increasing intervals (30 seconds to 16 hours) to *S. mansoni* miracidia.** Histograms show expression of the *BgHDAC* encoding transcript in snails at each time point from five biological replicates. Note the increase in fold change in the resistant BS-90 compared to the susceptible BBO2 snails after parasite infection. Significant expression normalized against expression of the myoglobin encoding transcript was measured by 2-way ANOVA and is indicated by number of asterisks on each histogram where ****, indicates the most significant value $p \leq 0.0001$, *** $p \leq 0.001$, ** $p \leq 0.01$, * $p \leq 0.05$, ns $p > 0.05$.

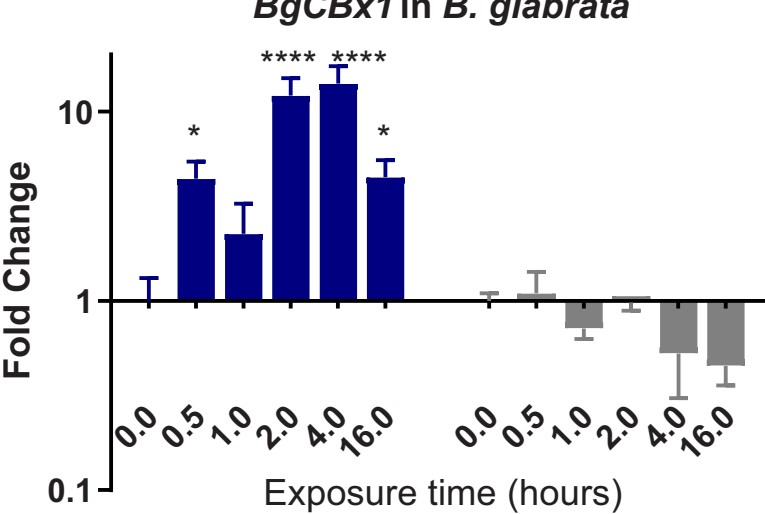

**Fig 3. qPCR analysis of RNA from either resistant BS-90 (blue histogram) or susceptible BBO2 (gray histograms) juvenile snails unexposed (0) or exposed for increasing intervals (30 seconds to 16 hours) *S. mansoni* miracidia.** Histograms show expression of the *BgCBx* encoding transcript in snails at each time point from five biological replicates. Note the increase in fold change in the resistant BS-90 compared to the susceptible BBO2 snails after parasite infection. Significant expression normalized against expression of the myoglobin encoding transcript was measured by 2-way ANOVA and is indicated by number of asterisks on each histogram where ****, indicates the most significant value $p \leq 0.0001$, *** $p \leq 0.001$, ** $p \leq 0.01$, * $p \leq 0.05$, ns $p > 0.05$.

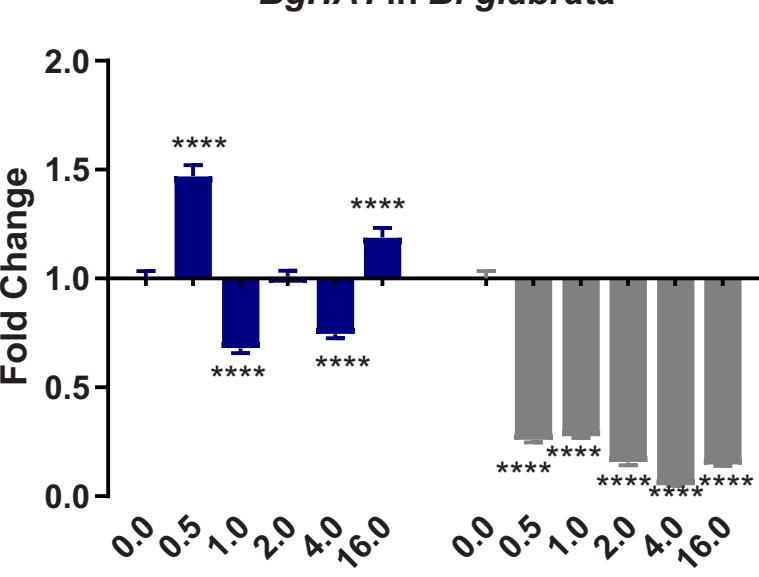

**Fig 4. qPCR analysis of RNA from either resistant BS-90 (blue histogram) or susceptible BBO2 (gray histograms) juvenile snails unexposed (0) or exposed for increasing intervals (30 seconds to 16 hours) to *S. mansoni* miracidia.** Histograms show expression of the *BgHAT* encoding transcript in snails at each time point from five biological replicates. Note the increase in fold change in the resistant BS-90 compared to the susceptible BBO2 snails after parasite infection. Significant expression normalized against expression of the myoglobin encoding transcript was measured by 2-way ANOVA and is indicated by number of asterisks on each histogram where ****, indicates the most significant value $p \leq 0.0001$, *** $p \leq 0.001$, ** $p \leq 0.01$, * $p \leq 0.05$, ns $p > 0.05$.

this (*BgPiwi*) transcript in the resistant (25˚C) and the susceptible juvenile (32˚C) BS-90 snails, with and without *S. mansoni* infection. The findings presented in Fig 7 demonstrates that unlike resistant BS90 snails, cultured at 25˚C, where the expression of the *BgPiwi* was upregulated following (2 hour) *S. mansoni* infection; in contrast, in susceptible juvenile BS-90 snails cultured at 32˚C, the expression of the *BgPiwi* transcript was downregulated as was shown above in the *S. mansoni* exposed BBO2 susceptible snail (Fig 1). Based on these data and given the known role of PIWI in silencing endogenous retrotransposable elements and previous results that revealed expression (upregulation) of the RT domain of *nimbus* occurs in juvenile susceptible but not resistant snails in response to *S. mansoni*, the functional role of PIWI in the epigenetics of *B. glabrata*/*S. mansoni* susceptibility was investigated by silencing the expression of the transcript encoding *BgPiwi* with corresponding siRNAs. As shown in Fig 8, investigation of the expression of the transcript encoding *BgPiwi* in siRNA/PEI transfected snails with (2 hours) and without (0 hours) *S. mansoni* exposure showed the knock-down of the *BgPiwi* encoding transcript in snails transfected with siRNAs corresponding to *BgPiwi*, but not to the universal mock siRNA. Use of *BgPiwi* dsRNA/PEI complexes instead of siRNA/PEI complexes to transfect BS-90 snails, similarly, produced the knock-down of the PIWI encoding transcript as observed with using siRNAs. To determine the biological effect of silencing *BgPiwi* in relation to *S. mansoni* infection in *BgPiwi* siRNA/PEI transfected BS90 snails, schistosome exposed transfected, untransfected snails, and Universal mock siRNA/PEI transfected snails were left at room temperature and evaluated at 4- and 6-weeks post-exposure (S2 and S3 Figs).

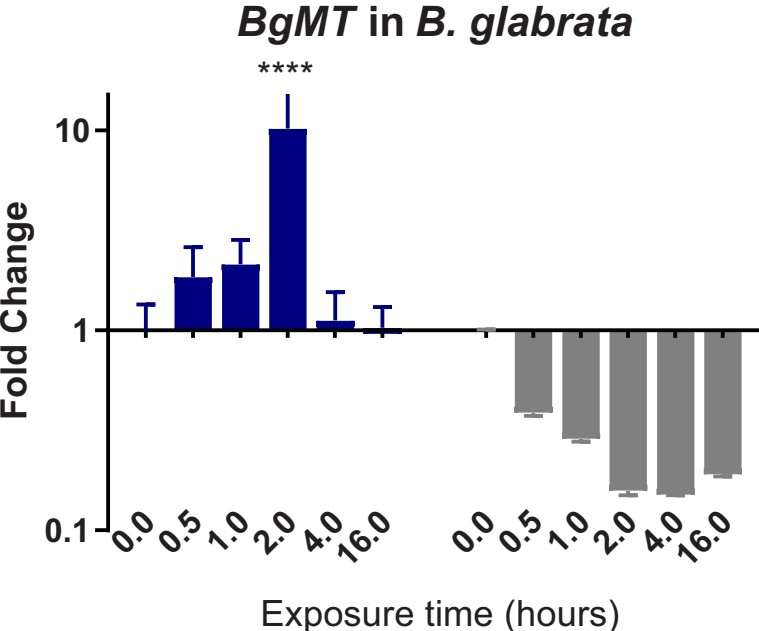

**Fig 5. qPCR analysis of RNA from either resistant BS-90 (blue histogram) or susceptible BBO2 (gray histograms) juvenile snails unexposed (0) or exposed for increasing intervals (30 seconds to 16 hours) to *S. mansoni* miracidia.** Histograms show expression of the *BgMT* encoding transcript in snails at each time point from five biological replicates. Note the increase in fold change in the resistant BS-90 compared to the susceptible BBO2 snails after parasite infection. Significant expression normalized against expression of the myoglobin encoding transcript was measured by 2-way ANOVA and is indicated by number of asterisks on each histogram where ****, indicates the most significant value $p \leq 0.0001$, *** $p \leq 0.001$, ** $p \leq 0.01$, * $p \leq 0.05$, ns $p > 0.05$.

## Knock-down of PIWI encoding transcript with *siBgPiwi*/PEI complexes concurrently upregulates the *nimbus* RT in transfected BS90 snails

Because PIWI suppresses the expression of retrotransposable elements, such as the *B. glabrata* non-LTR retrotransposable element *nimbus*, the transcription of the RT domain of this element

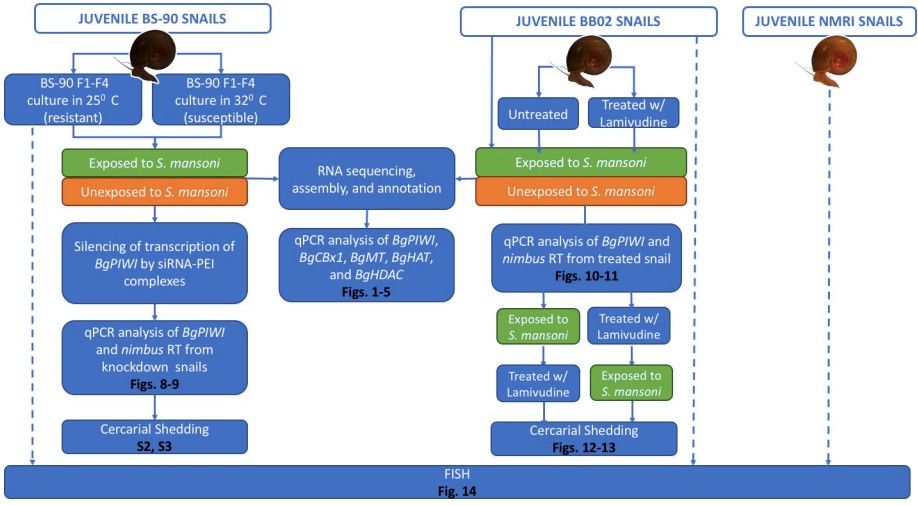

**Fig 6. Graphical flow chart illustration of the results and experimental design.** Juvenile snails were used in the study and were exposed to *S. mansoni* miracidia as described on MATERIALS and METHODS.

**Table 3. The amino acid sequence of *BgPiwi* isoform 1 is conserved in comparison to those from other organisms.**

| | Query Cover | E-value | Percent Identity | Accession |
|---|---|---|---|---|
| piwi-like protein 1 (*Danio rerio*) | 93% | 0.0 | 42.14% | NP_899181.1 |
| piwi-like protein 1, isoform 1 (*Homo sapiens*) | 93% | 0.0 | 42.97% | NP_004755.2 |
| piwi-like protein 1 (*Mus musculus*) | 92% | 0.0 | 42.28% | NP_067286.1 |
| argonaute 3, isoform D (*Drosophila melanogaster*) | 85% | 0.0 | 39.11% | NP_001036627.2 |
| piwi-like protein (*Caenorhabditis elegans*) | 86% | 4e-162 | 33.79% | NP_492121.1 |

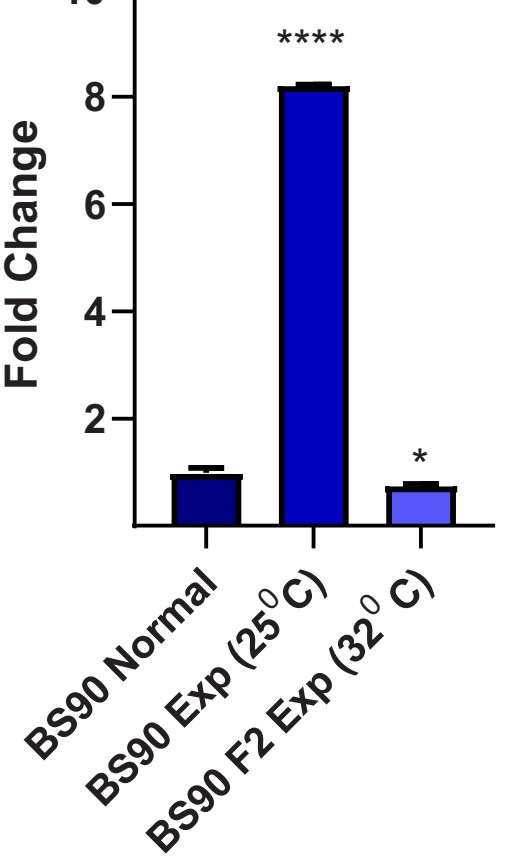

**Fig 7. qPCR analysis of RNA from either non permissive (25°C) resistant BS-90 (blue histogram) or permissive (32°C) susceptible BS-90 (gray histograms) juvenile snails unexposed (normal) or exposed for 2hr to *S. mansoni* miracidia.** Histograms show expression of the *BgPiwi* encoding transcript in these snails residing at different temperatures. Note the significant induction (8-fold change) in 25°C non-permissive BS-90 snails compared the down regulation of the transcript in permissive BS-90 snails residing at 32°C after parasite infection. Fold change was determined as described in MATERIALS and METHODS. Significant expression normalized against expression of the myoglobin encoding transcript was measured by 2-way ANOVA and is indicated by number of asterixis on each histogram where ****, indicates the most significant value $p \leq 0.0001$, *** $p \leq 0.001$, ** $p \leq 0.01$, * $p \leq 0.05$, ns $p > 0.05$.

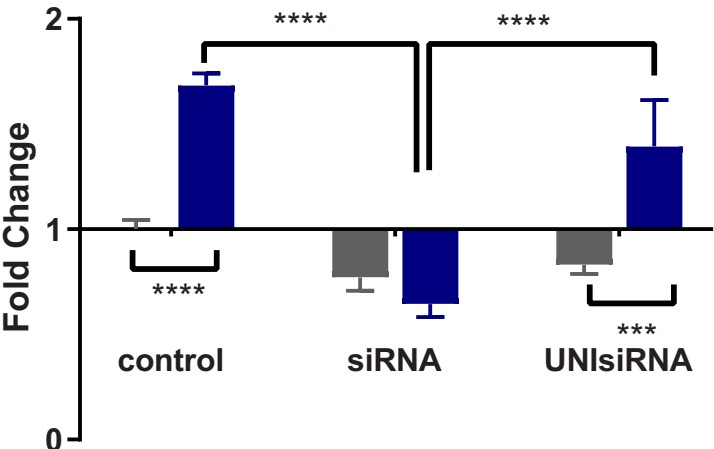

**Fig 8. qPCR analysis of RNA from resistant BS-90 juvenile snails unexposed (gray) or exposed (blue) for 2hr to *S. mansoni* miracidia.** Histograms show expression of the *BgPiwi* encoding transcript in normal BS-90 snails (control) or those transfected with *BgPiwi* siRNA. Note induction of the *BgPiwi* encoding transcript occurs in *S. mansoni* exposed control BS-90 snails and the knock down of the transcript in BS-90 (exposed and unexposed) snails transfected with *BgPiwi* siRNA. In BS-90 snails transfected with mock UNIsiRNA, note the upregulation of the *BgPiwi* encoding transcript in exposed snails similar to induction observed in control exposed snails. Fold change was determined as described in MATERIALS and METHODS. Significant expression normalized against expression of the myoglobin encoding transcript was measured by 2-way ANOVA and is indicated by number of asterixis on each histogram where ****, indicates the most significant value $p \leq 0.0001$, *** $p \leq 0.001$, ** $p \leq 0.01$, * $p \leq 0.05$, ns $p > 0.05$.

was investigated in either unexposed (0) or *S. mansoni* (2 hour) exposed BS-90 snails that were transfected with either *BgPiwi* siRNA or mock universal siRNA. The same cDNA templates utilized in the qPCR assays shown in Fig 8 were utilized in the analysis of the expression of *nimbus* RT (Fig 9). Using gene specific primers corresponding to the RT domain of *nimbus*, results showed that in untransfected BS-90 snails that were unexposed (0) to *S. mansoni*, that similar to resistant BS-90 transfected with mock siRNA (UnisiRNA/PEI), the expression of *nimbus* RT remained low upon infection of the normal resistant BS-90 snails as previously demonstrated [18]. Furthermore, in BS-90 snails transfected with *BgPiwi* siRNA before exposure, the *nimbus* RT encoding transcript was upregulated while *BgPiwi* transcript was silenced as expected.

### Lamivudine RT inhibitor treatment of susceptible BBO2 snails differentially regulates transcription of *nimbus* RT and *BgPiwi*

To further examine the interplay between the expression of *BgPiwi* and *nimbus* RT encoding transcripts in relation to *S. mansoni* infection of *B. glabrata*. The susceptible juvenile BBO2 snail was treated with the RT inhibitor, lamivudine, as described in MATERIALS and METHODS prior to exposure to *S. mansoni*. As shown in Figs 10 and 11, qPCR analysis of the same cDNA template prepared from BBO2 susceptible snails that were either treated or untreated with different concentrations (100 ng/ml and 200 ng/ml) of lamivudine before exposure (for 2 hours) or not exposed (0 hour) to *S. mansoni* were performed by utilizing primers corresponding to the *BgPiwi* (Fig 10) or *nimbus* RT (Fig 11) encoding transcripts. Fig 10 shows the expression of *BgPiwi* remained relatively unchanged with drug treatment while that of *nimbus* RT (Fig 11) was downregulated with Lamivudine. Both concentrations of lamivudine, either 100 or 200 ng/ml, used to treat BBO2 snails prior to exposure suppressed the expression of the *nimbus* RT encoding transcript in lamivudine–treated snails.

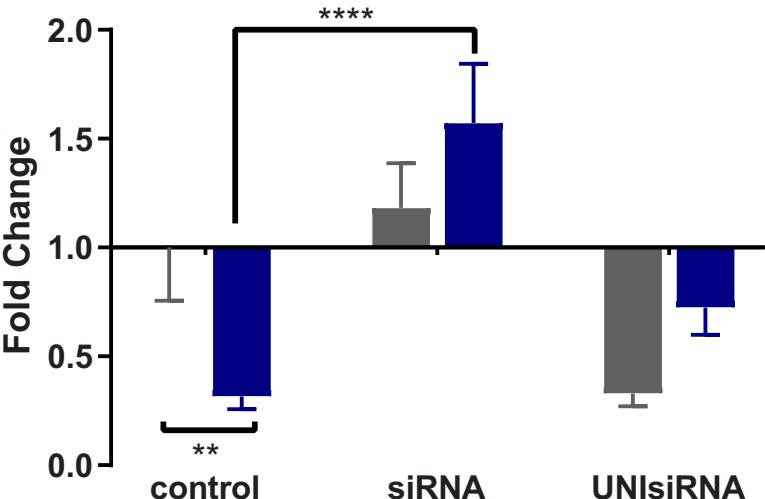

**Fig 9. qPCR analysis of RNA from resistant BS-90 juvenile snails unexposed (gray) or exposed (blue) for 2hr to *S. mansoni* miracidia.** Histograms show expression of the *nimbusRT* encoding transcript in normal BS-90 snails (control) or those transfected with *BgPiwi* siRNA. Note the down regulation of the *nimbusRT* encoding transcript in *S. mansoni* exposed control BS-90 snails and the upregulation of *nimbusRT* transcript in BS-90 (exposed and unexposed) snails transfected with *BgPiwi* siRNA where transcript encoding *BgPiwi* has been knocked-down (shown in Fig 2B). In BS-90 snails transfected with mock UNIsiRNA, note the down regulation of the *nimbusRT* encoding transcript in exposed snails similar to that observed in control exposed snails. Fold change was determined as described in MATERIALS and METHODS. Significant expression normalized against expression of the myoglobin encoding transcript was measured by 2-way ANOVA and is indicated by number of asterixis on each histogram where ****, indicates the most significant value $p \leq 0.0001$, *** $p \leq 0.001$, ** $p \leq 0.01$, * $p \leq 0.05$, ns $p > 0.05$.

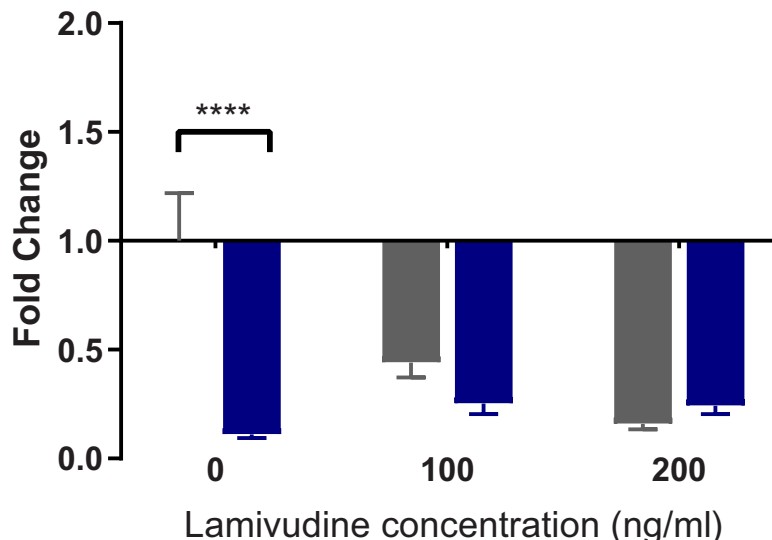

**Fig 10. qPCR analysis of RNA from susceptible BBO2 juvenile snails unexposed (gray) or exposed (blue) for 2hr to *S. mansoni* miracidia.** Histograms show expression of the *BgPiwi* encoding transcript in normal BBO2 snails (0) or those treated with RT inhibitor Lamivudine (100 ng/ml or 200ng/m1). Note the down regulation of the *BgPiwi* encoding transcript in exposed control (0) snails and lamivudine treated snails (exposed and unexposed) snails. Fold change was determined as described in MATERIALS and METHODS. Significant expression normalized against expression of the myoglobin encoding transcript was measured by 2-way ANOVA and is indicated by number of asterixis on each histogram where ****, indicates the most significant value $p \leq 0.0001$, *** $p \leq 0.001$, ** $p \leq 0.01$, * $p \leq 0.05$, ns $p > 0.05$.

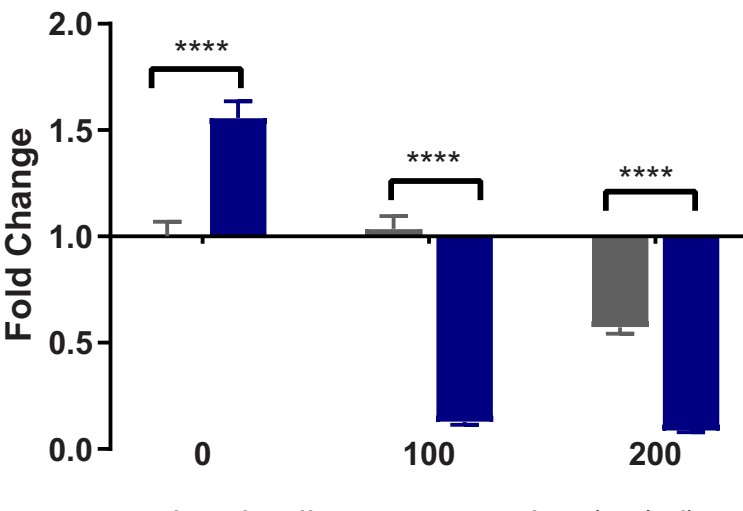

**Fig 11. qPCR analysis of RNA from susceptible BBO2 juvenile snails unexposed (gray) or exposed (blue) for 2hr to *S. mansoni* miracidia.** Histograms show expression of the *nimbusRT* encoding transcript in normal BBO2 snails (0) or those treated with RT inhibitor, lamivudine (100 ng/ml or 200 ng/m1). Note the upregulation of the *nimbusRT* encoding transcript in exposed control (0) snails and down regulation of this transcript in lamivudine-treated snails (exposed and unexposed) snails. Fold change was determined as described in MATERIALS and METHODS. Significant expression normalized against expression of the myoglobin encoding transcript was measured by 2-way ANOVA and is indicated by number of asterisks on each histogram where ****, indicates the most significant value $p \leq 0.0001$, *** $p \leq 0.001$, ** $p \leq 0.01$, * $p \leq 0.05$, ns $p > 0.05$.

## Lamivudine blocks schistosome infection of BBO2 snails

Because the silencing of the *BgPiwi* encoding transcript rendered the resistant BS-90 snail susceptible, concurrent with the down- and up-regulation of the *BgPiwi* encoding transcript and *nimbus* RT encoding transcripts, respectively, and since follow-up qPCR analysis using the same cDNA templates showed that *nimbus* RT was knocked down by lamivudine treatment of the susceptible BBO2 snail, the effect of this drug on the ability of the susceptible snail to sustain a viable infection was examined (Fig 6). As shown in Figs 12 and 13, where juvenile BBO2 snails were treated with 100 ng/ml of lamivudine either before or two weeks after exposure to *S. mansoni* miracidia, snails failed to shed cercariae when treated with lamivudine prior to exposure (Fig 12). However, snails treated with BPPA (anthraquinone diacetate), another RT inhibitor that specifically blocks the reverse transcriptase activity of the human homolog of telomerase (hTERT) in *B. glabrata*, shed cercariae when treated before exposure but not when treated at 14 days post *S. mansoni* miracidia exposure (Fig 13). In contrast to these results, the same delayed treatment regimen with lamivudine (2 weeks post- exposure) failed to block infection. The BBO2 snails failed to shed cercariae at 10 weeks post exposure only when pretreated with lamivudine prior to exposure. These results showed that early induction of *nimbus* RT in the snail host following schistosome infection is important for parasite survival.

## Relocalization of the *PIWI* gene locus occurs in resistant *B. glabrata* snails upon infection

We had demonstrated that co-culture of snail Bge cells with live parasite [34,36], and infection of whole snails with live parasite, both lead to non-random gene loci relocation within the

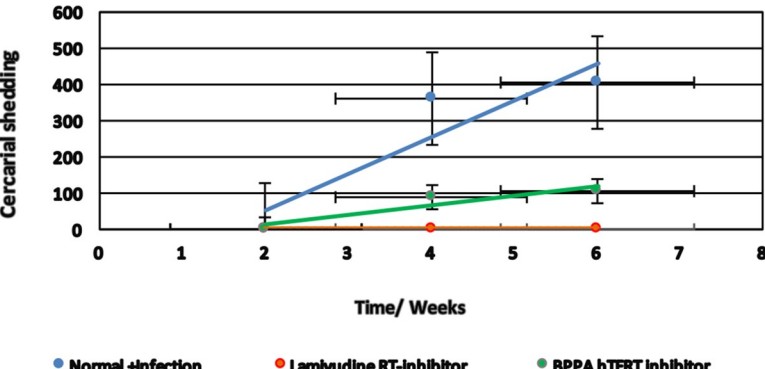

**Fig 12. To determine the effect of lamivudine in relation to *S. mansoni* infection of susceptible BBO2 snails, snails were treated before exposure with 100 ng/ml of the RT inhibitor, maintained at room temperature, and evaluated for up to 6 weeks post-exposure.** Note that the BBO2 snails treated before infection with lamivudine failed to shed cercariae at 6 weeks post-exposure to *S. mansoni* unlike in untreated (control) snails. Also note that BBO2 snails treated with the hTERT RT inhibitor BBPA before exposure, unlike Lamivudine shed cercariae at 6 weeks post-exposure.

interphase nuclei of the host snail cells, correlated with gene up-regulation [13,19,40], with notable differences in gene movement between susceptible and resistant snails. We have been able to demonstrate again that gene loci change their non-random nuclear location with changes in gene expression. A FISH probe containing the sequences for *B. glabrata PIWI* was employed to delineate the nuclear position of the *piwi* gene loci in three snail strains, BS-90 (resistant) and the two susceptible strains, BB02 and NMRI (Fig 14A–14F). The nuclear positioning of the gene loci were analyzed using the erosion analysis script for gene and chromosome positioning, we have used previously for different species [41–43] and can be assigned to a peripheral (Fig 14A), intermediate (Fig 14B) or internal (Fig 14C) location in cell nuclei.

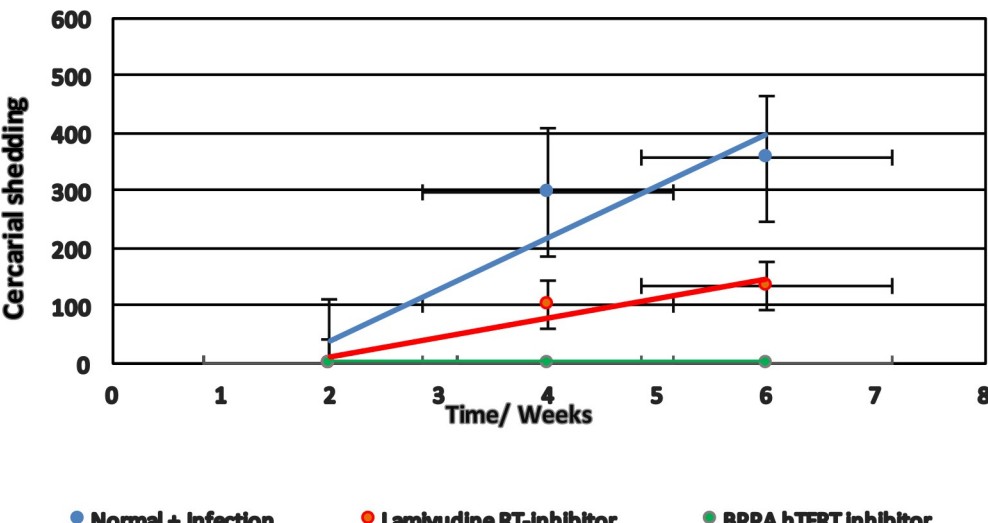

**Fig 13. The effect of treating BBO2 susceptible snails with 100 ng/ml of lamivudine at 14 days post -exposure to *S. mansoni* was compared to the effect of treating susceptible BBO2 snails before exposure with 100 ng/ml of BPPA as described in MATERIALS and METHODS and left at room temperature and evaluated for up to 6 weeks post-exposure.** Note that the BBO2 snails treated before *S. mansoni* exposure with 100 ng BPPA failed to shed cercariae at 6- weeks post- exposure unlike in untreated (control) snails. Also note that BBO2 snails treated with lamivudine at 14 days after infection shed cercariae at 6 weeks post-exposure.

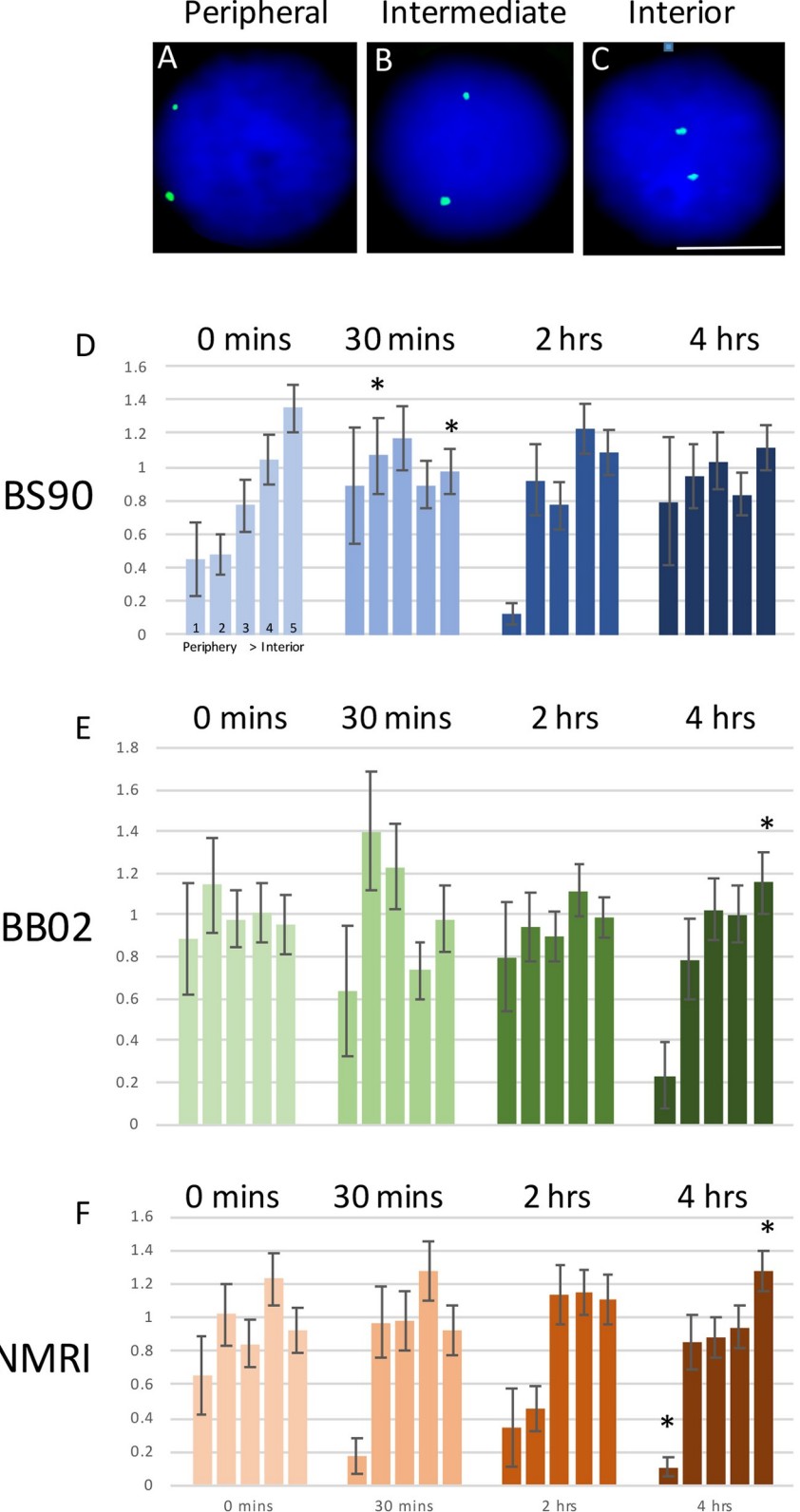

**Fig 14. Nuclei, in blue, as they are stained with DAPI, were isolated from snail strains BS90, BB02 and NIMR ovo-testis and subjected to 2D-fluorescence in situ hybridisation (FISH) with fluorescein labelled probes for the transposable element,** *piwi* **(green).** Panel A displays a peripheral nuclear location, panel B an intermediate nuclear

location and panel C a location at the nuclear interior for the two gene loci. Scale bar = 5 μm. Using a bespoke nuclear positioning script that creates five concentric shells of equal area, shells 1–5, with shell 1 being the nuclear periphery and shell 5 the nuclear center, the percentage of fluorescent green gene signal is measured in each shell for over 50 nuclei and divided by the percentage of blue fluorescence signals for the DNA content (DAPI) in each shell for normalization. The data are averaged and plotted as bar charts with standard error of the mean (SEM) as error bars. These graphs are displayed in panels D-F, showing the distribution of the normalized *piwi* gene signals from the nuclear periphery (shell 1) to the nuclear interior (shell 5), 30 mins, 2 hours and 4 hours after an infection with *Schistosoma mansoni* at 0 hours. Panel D displays data from BS90 snails, panel E data from BB02 snails and panel F from NMRI snails.

Notably, the *PIWI* gene signal is in different nuclear compartments for the resistant and susceptible snails. Indeed, in BS-90 (Fig 14D), the gene loci are found towards the nuclear interior and upon infection there is a relocation towards the nuclear periphery at 30 minutes after infection (Fig 7D), which coincides with the increase in *PIWI* transcripts (Fig 1). By two hours, the gene loci are relocating back to the nuclear interior (Fig 14D).

The two susceptible snail strains of BB02 (Fig 14E) and NMRI (Fig 14F) display a similar gene loci location for *PIWI*, an intermediate location, different from the resistant BS-90 strain (Fig 14D). Furthermore, the *PIWI* gene locus does not change location until 4 hours post-infection, where in both snail strains, the gene locus is in the nuclear interior, a location has been correlated with down-regulation of expression in BS-90, and in BB02 (Fig 1). Together, these findings further support the notion that the parasite is able to influence genome behavior within its host for its own advantage in an epigenetic mechanism, through functional spatial positioning.

## Discussion

Significant progress has been made in recent years towards elucidating the molecular basis of *S. mansoni* resistance/susceptibility in the intermediate snail vector *B. glabrata*. From these studies, it has become clear that the snail and schistosome relationship is complex and highly variable [13,20,44]. This study was undertaken to determine the role of epigenetics in shaping the relationship between the snail and the schistosome. The genetics of this interaction is well known and several molecular determinants that underlie the innate defense anti-schistosome response in *B. glabrata* have been identified as have sequences that are linked in the snail genome to resistance [14,21,45]. Epigenetics describes the inheritance of a reversible phenotype that is not influenced by any change in the sequence of DNA. Transgenerational epigenetic inheritance induced by environmental changes have recently reported the role of PIWI and small piRNA in stress-induced genome modifications (see [46]). The impact of viral infection on the siRNA and Argonaute /PIWI pathway has mainly been reported in insects [47,48]. However, little is known regarding genes and proteins involved in this insect Ago-2/RNAi antiviral/stress defense response [49].

Variation in *B. glabrata* susceptibility to *S. mansoni* was investigated by examining the regulation of key transcripts that play a role in epigenetics. Our approach was to use representative juvenile snail stocks that are either resistant (BS-90) or susceptible (BBO2) to the NMRI strain of *S. mansoni*, to examine the temporal regulation of transcripts encoding PIWI (*BgPiwi*), chromobox protein homolog 1 (*BgCBx1*), histone acetyl transferase (*BgHAT*), histone deacetylase (*BgHDAC*), and metallotransferase (*BgMT*) in these snail stocks within 30 min to 16hr post- exposure to *S. mansoni*. The differential expression of these transcripts was initially identified by comparing RNA-seq datasets that were generated from non-permissive juvenile BS-90 snails, cultured at room temperature (25˚C), and their permissive counterparts cultured for two generations at 32˚C. The snail infections were carried out exclusively with juvenile snails.

Thus, to further validate the differential expression of these transcripts, resistant juvenile BS-90 and susceptible juvenile BBO2 snails were exposed to *S. mansoni* before using real-time qPCR to confirm their modulation pre- and post- exposure to *S. mansoni*. Transcripts encoding PIWI (*BgPiwi*), chromobox protein homolog 1 (*BgCBx1*), histone acetyl transferase (*BgHAT*), histone deacetylase (*BgHDAC*) and metallotransferase (*BgMT*) were upregulated (1.8 to 10-fold) in the resistant (BS-90) snail as compared to their downregulation in the susceptible juvenile snail (BBO2). Upregulation of the majority of these transcripts occurred within the first 30 min of exposure to *S. mansoni*, peaking at 120 min before subsiding. In earlier reports, we showed that the regulation of the RT domain of the *B. glabrata* endogenous non-LTR-retrotransposable element, *nimbus*, was linked to the early differential stress response observed between juvenile resistant and susceptible snails [18]. Accordingly, induction of RT occurred concurrently with the upregulation of the transcript encoding Hsp70 in the susceptible but not the resistant snail after *S. mansoni* infection. Exposure of *B. glabrata* to irradiated attenuated miracidia, however, failed to induce these stress-related transcripts in early-infected juvenile susceptible snails [25]. Given those findings and results presented here showing that upregulation of the *B. glabrata PIWI* encoding transcript, *BgPiwi*, occurs in resistant BS-90 snails residing at room temperature (where they are resistant) but not in their susceptible counterparts residing at 32˚C, we examined the expression of the *BgPiwi* transcript more closely in relation to the early expression of *nimbus* RT in susceptible and resistant snails. To reiterate, changes in transcription regulation were evident within 30 minutes of infection. In this regard, these novel findings differ from those described by others where molecular interactions between the snail and schistosomes were examined much later after miracidia penetration at which time the responses we now report might have waned.

The existence of several piRNA sequences in *B. glabrata* has been described but their role in a PIWI-piRNA mediated anti-parasite defense mechanism in the snail remains elusive [50]. However, to determine whether a PIWI gene-silencing mechanism that involves *nimbus* RT plays a role in blocking transmission of schistosomes in *B. glabrata*, we utilized a previously developed PEI-mediated soaking method to deliver two different *BgPiwi* corresponding duplex siRNAs, simultaneously, into the resistant BS-90 snail, thereby knocking-down the expression of the PIWI encoding transcript. The RNAi suppression of PIWI transcription, rendered these *BgPiwi* siRNA/PEI transfected resistant BS-90 snails susceptible and thus able to shed cercariae (S2 and S3 Figs). While the transcript encoding *BgPiwi* was reduced in siRNA/PEI transfected snails, in contrast, the expression of *nimbus* RT-encoding transcript was upregulated, indicating that a gene silencing mechanism mediated by the interplay of *BgPiwi* and modulation of the transcription of *nimbus* RT plays a major role in *B. glabrata* susceptibility to *S. mansoni*. To provide further support for these findings, the susceptible BBO2 snail was treated with lamivudine, a known RT inhibitor. Lamivudine is a RT nucleoside analog inhibitor that is used to treat hepatitis B and HIV/ AIDS [51]. Treatment of either susceptible BBO2 or NMRI snails prior to schistosome exposure, consistently, after several biological replicates blocked *S. mansoni* infection in the snail. By contrast, BPPA, another RT inhibitor that specifically targets the RT activity of telomerase, did not block infection in *B. glabrata* by *S. mansoni* when snails were treated before parasite exposure. However, in snails treated at 2 weeks after exposure, BPPA prevented infection unlike what was observed with this treatment regimen with lamivudine. These findings indicated that the mechanism of action of this *nimbus*/PIWI interplay occurs very early in the *S. mansoni* and *B. glabrata* interaction. In ongoing studies, we have shown that an hTERT homolog is absent in the *S. mansoni* genome with the snail ortholog showing significant identity at the amino acid level to the human enzyme. Interestingly, two PIWI related protein isoforms (1and 2) were found to occur in *B. glabrata*. The

focus of the present study was on *B. glabrata* PIWI isoform 1 which was found to be highly conserved at the amino acid level to both vertebrate and invertebrate PIWI homologs.

Work is currently underway to determine if using the same siRNA mediated gene silencing strategy as described above will reveal the significance of the other transcripts identified in this study and their functional role in epigenetics of the *S. mansoni/B. glabrata* relationship. Previously, we showed that hypomethylation of the stress Hsp70 protein locus precedes the early upregulation of the Hsp70 encoding transcript in *S. mansoni* exposed susceptible (NMRI) but not resistant (BS-90) juvenile snails [52]. The findings here that the transcript encoding MT was upregulated in juvenile BS-90 resistant but not susceptible BBO2 upon early parasite infection snails supports this earlier result [52]. Ideal follow up experiments to further verify the involvement of all the transcripts identified in this study in epigenetics of snail plasticity to *S. mansoni* susceptibility will be to edit their CDS by a permanent gene mutation such as by CRISPR/Cas9 to knockout their function. These approaches can likely contribute to deciphering epigenetic processes that underlie the susceptibility of the snail to the parasite. Toward this objective, molecular toolkits for gene editing in *B. glabrata* are being developed. CRISPR gene editing has been used to edit the allograft inflammatory factor gene in the *Bge* embryonic cell line from *B. glabrata* [53] and is finding utility in editing the schistosome genome [54,55]. Confirming earlier findings, we show that in intact juvenile resistant and susceptible snails, within a short period post-exposure to *S. mansoni*, the non-random movement of the *PIWI* locus within interphase nuclei in relation to its active transcription depending on the susceptibility phenotype of the snail [19]. We have shown for the schistosome mediated relocation of gene loci that movement in interphase nuclei (from peripheral to interior location) occurs early after infection of susceptible, not resistant snails, and precedes transcription of the gene loci in question. A soluble factor(s) within excretory secretory products (ESPs) from the wild-type miracidia that mediates the systemic reorganization of the host genome is yet to be uncovered. Aside from schistosomes, viruses are the only other pathogens that have been shown to also mediate non-random gene relocation. However, schistosomes are the first metazoan parasites that have been shown to possess the ability to manipulate the genome of the host in this profound spatio-epigenetic fashion.

To conclude, although a molecular basis exists for *B. glabrata* susceptibility to *S. mansoni*, it remains far from clear what precise pathways/mechanisms are responsible for parasite survival or rejection in the early infected snail. This is the first study to show that epigenetics, involving the interplay of PIWI and the endogenous non LTR-retrotransposable *nimbus*, plays a role in the plasticity of snail susceptibility to *S. mansoni*.

## Supporting information

**S1 Fig. Title, Phylogenetic tree constructed from amino acid sequences of *BgPiwi* and those from other organisms.** Phylogenetic tree constructed using the neighbor-joining method from the amino acid sequences of *BgPiwi* isoform 1 and of other organisms (Table 3). The data reveal that the *B. glabrata* isoform 1 protein shows closer evolutionary relationship to the *C. elegans*, human, mouse, and zebra fish amino acid sequences than to *D. melanogaster*. (TIF)

**S2 Fig. *BgPiwi* siRNA transfected BS-90 snail shedding cercariae at 4 weeks post-exposure to *S. mansoni*.** To determine the biological effect of silencing *BgPiwi* in relation to *S. mansoni* infection in *BgPiwi* siRNA/PEI transfected BS-90 snails, schistosome exposed *BgPiwi* siRNA transfected and untransfected snails were left at room temperature and evaluated at 4- and 6-weeks post-exposure. Note that the BS-90 snail transfected with *BgPiwi* siRNA shed cercariae

at 4 weeks post-exposure to *S*. *mansoni*.
(TIF)

**S3 Fig.** *BgPiwi* **siRNA transfected BS-90 snail shedding cercariae at 6 weeks post-exposure to** *S*. *mansoni*. To determine the biological effect of silencing *BgPiwi* in relation to *S*. *mansoni* infection in *BgPiwi* siRNA/PEI transfected BS-90 snails, schistosome exposed *BgPiwi* siRNA transfected and untransfected snails were left at room temperature and evaluated at 4- and 6-weeks post-exposure. Note that the BS-90 snail transfected with *BgPiwi* siRNA shed cercariae at 6 weeks post-exposure to *S*. *mansoni*.
(TIF)

## Acknowledgments

We thank Ms. Oumsalama Elhelu, Dr. Ruth Joy Relador, Mr. Adeola Fagunloye and Ms. Rita Abalada Pedrosa Torres Pereira for their technical help. We also thank Dean April Massey for her support in allowing Dr. Michael Smith to conduct his research for his Ph.D. dissertation at UDC. We thank Dr. Margaret Mentink- Kane at the NIAID Schistosomiasis Resource Center of the Biomedical Research Institute for the parasite material used in this study.

## Author Contributions

**Conceptualization:** Joanna M. Bridger, Matty Knight.

**Data curation:** Nashwah Alsultan, Andrea Borns, Najib M. El-Sayed, Matty Knight.

**Formal analysis:** Olayemi G. Fagunloye, Nana Adjoa Pels, Daniel A. Horton, Andrea Borns, Freddie Dixon, Victoria H. Mann, Clarence Lee, Najib M. El-Sayed, Joanna M. Bridger, Matty Knight.

**Funding acquisition:** Matty Knight.

**Investigation:** Michael Smith, Swara Yadav, Olayemi G. Fagunloye, Nana Adjoa Pels, Daniel A. Horton, Nashwah Alsultan, Andrea Borns, Victoria H. Mann, Paul J. Brindley, Najib M. El-Sayed, Joanna M. Bridger, Matty Knight.

**Methodology:** Michael Smith, Swara Yadav, Olayemi G. Fagunloye, Nana Adjoa Pels, Daniel A. Horton, Nashwah Alsultan, Andrea Borns, Victoria H. Mann, Paul J. Brindley, Najib M. El-Sayed, Matty Knight.

**Project administration:** Carolyn Cousin, Clarence Lee, Joanna M. Bridger, Matty Knight.

**Resources:** Freddie Dixon, Matty Knight.

**Software:** Michael Smith, Freddie Dixon, Najib M. El-Sayed.

**Supervision:** Paul J. Brindley, Joanna M. Bridger, Matty Knight.

**Validation:** Michael Smith, Swara Yadav, Olayemi G. Fagunloye, Nana Adjoa Pels, Daniel A. Horton, Nashwah Alsultan, Najib M. El-Sayed, Joanna M. Bridger, Matty Knight.

**Visualization:** Michael Smith.

**Writing – original draft:** Paul J. Brindley, Joanna M. Bridger, Matty Knight.

**Writing – review & editing:** Michael Smith, Daniel A. Horton, Carolyn Cousin, Freddie Dixon, Victoria H. Mann, Clarence Lee, Paul J. Brindley, Najib M. El-Sayed, Joanna M. Bridger, Matty Knight.

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
