## [Decision Letter · Decision Letter 0]

15 Feb 2021

Dear Dr. Knight,

Thank you very much for submitting your manuscript "PIWI silencing mechanism involving the retrotransposon nimbus orchestrates resistance to infection with Schistosoma mansoni in the snail vector, Biomphalaria glabrata" for consideration at PLOS Neglected Tropical Diseases. As with all papers reviewed by the journal, your manuscript was reviewed by members of the editorial board and by several independent reviewers. In light of the reviews (below this email), we would like to invite the resubmission of a significantly-revised version that takes into account the reviewers' comments. 

We cannot make any decision about publication until we have seen the revised manuscript and your response to the reviewers' comments. Your revised manuscript is also likely to be sent to reviewers for further evaluation.

Sincerely,

Philip T. LoVerde

Guest Editor

Michael Hsieh

Deputy Editor

Reviewer's Responses to Questions

**Key Review Criteria Required for Acceptance?**

**Methods**

-Are the objectives of the study clearly articulated with a clear testable hypothesis stated?

-Is the study design appropriate to address the stated objectives?

-Is the population clearly described and appropriate for the hypothesis being tested?

-Is the sample size sufficient to ensure adequate power to address the hypothesis being tested?

-Were correct statistical analysis used to support conclusions?

-Are there concerns about ethical or regulatory requirements being met?

Reviewer #1: I do not believe that new analyses are required for acceptance but the manuscript should be comprehensively revised.

Reviewer #2: -Are the objectives of the study clearly articulated with a clear testable hypothesis stated? y

-Is the study design appropriate to address the stated objectives? nd

-Is the population clearly described and appropriate for the hypothesis being tested? y

-Is the sample size sufficient to ensure adequate power to address the hypothesis being tested? nd

-Were correct statistical analysis used to support conclusions? n

-Are there concerns about ethical or regulatory requirements being met? nd, it may be included to be published

**Results**

-Does the analysis presented match the analysis plan?

-Are the results clearly and completely presented?

-Are the figures (Tables, Images) of sufficient quality for clarity?

Reviewer #1: Figures were mostly clear. See comments on figure legends in attached document. Photographs of shedding snails doesn't appear to be necessary.

Reviewer #2: -Does the analysis presented match the analysis plan? y

-Are the results clearly and completely presented? y

-Are the figures (Tables, Images) of sufficient quality for clarity? y

**Conclusions**

-Are the conclusions supported by the data presented?

-Are the limitations of analysis clearly described?

-Do the authors discuss how these data can be helpful to advance our understanding of the topic under study?

-Is public health relevance addressed?

Reviewer #1: (No Response)

Reviewer #2: -Does the analysis presented match the analysis plan? y

-Are the results clearly and completely presented?y

-Are the figures (Tables, Images) of sufficient quality for clarity?y

**Editorial and Data Presentation Modifications?**

Reviewer #1: (No Response)

Reviewer #2: (No Response)

**Summary and General Comments**

Reviewer #1: Please see attached document. I selected accept with minor revisions as I do not think any additional experiments need to be carried out. However, I believe the manuscript should be comprehensively revised.

This manuscript describes the findings of a comprehensive study on the interplay of piwi and nimbus, and their potential role in the resistance or susceptibility of Biomphalaria snails to schistosome infection. This research is original and important to the field. It will also be of interest to researchers in other fields such as epigenetics and invertebrate immunology. Overall I recommend the following major editing. I recommend the following points be addressed and I’ve listed suggested edits.

• Overall, there was variation in formatting and writing style, perhaps as there are multiple authors. I recommend revising the paper so it is more uniform.

• It would be helpful to clarify why the 2 different siRNAs were used together rather than separately. 

• Can the authors provide any explanation for why in Fig 3, UNI siRNA treated unexposed BS-90 showed a much greater decrease in nimbus expression than in UNIsiRNA treated and schistosome-exposed BS-90?

• This was a comprehensive study comprising several different methods/experiments. A flow chart or graphic giving overview of all methods would be helpful to readers. 

• Likewise, a graphic summarizing the results would be helpful.

• The description of methods used to acquire sequences in order to design primers for qPCR were confusing (lines 209-223). There was mention of exploring the phylogeny of the B. glabrata orthologs but it isn’t clear why this was performed and I didn’t see the resulting data anywhere.

• If space in journal is limited, Figures 2B and 2C do not appear to be absolutely necessary. A table with number cercariae shed might be more impactful.

• The significance of the use of the BPPA hTERT inhibitor, and that it stops cercarial shedding when snails treated with it 2 weeks post-infection, was not clear.

• It was not clear why the 2 siRNAs were used simultaneously not individually.

Reviewer #2: The manuscript entitled “PIWI silencing mechanism involving the retrotransposon nimbus orchestrates resistance to infection with Schistosoma mansoni in the snail vector, Biomphalaria glabrata” is focusing on to decipher the role of bgPIWI snail resistance to one of its parasite, S. mansoni. Firstly, the authors uses a comparative transcriptomic approach to analyze the differential expression genes between permissive and non per-issive BS90 snails, a model of choice to investigate the link between snail epigenetics and resistance. Then, by a classical QPCR approach , they confirmed the differential gene expression for BgPiwi, BgMTn BgCBx1, BgHAT et HDAC between BS90 (non permissive) and another snail strain BBO2 (or BB02?) which is susceptible to S. mansoni. siRNA experminents targeting bgPIWI render resistant BS90 snails susceptible. A correlation has been done between bgPIWI and a non LTR retrotransposable element nimbus expression. RT inhibition by lamivudine affect snail susceptibility against S. mansoni. 

This study could be of relevance for the field. However, I have major concerns regarding the method used and the interpretation of the results. These concerns are detailed below.

MAJOR COMMENTS

Statistics methods used could be improved. For instance, multiple time points (parasite challenge) could be compared globally using a Kruskal-Wallis or an ANOVA test (depending on normality and homoscedasticity of the distributions) and post-hoc test can be then performed to identify pairwise differences. No test on normality data was mentioned for all analyzed data. Please provide more details for the statistical analysis

The authors also do not report the validation of the primer couples they are using in terms of PCR efficiency; thus, it is not clear whether they meet the conditions required for DeltaDeltaCt use. Note there are also statistics that are specifically dedicated to qRT-PCR (Pfaffl method).

In the abstract, the authors mentioned that comparative analysis will be performed on permissive and non-permissive BS90 snails, and the strain BBO2 was not indicated. All in all, the reader might reasonably expect that all comparisons are made between permissive and not permissive BS90, but this is not the case. Why ? To my mind, it will reinforce the conclusions of the paper and it will be a best straight strategy to claim on the bgPiwi role. All results performed with the BBO2 strain could be extrapolated for the BS90 F2 (permissive condition) ? It will be elegant to reinforce the conclusion to have the results from qPCR analysis of RNA from permissive BS90 exposed to parasite. In the same way, treatment with inhibitors could be performed on permissive BS90 snails and compared to non-permissive snails.

Please for all QPCR analysis, give the same scale for the y-axis. For example in Fig2A, bgpiwi expression seems to be unchanged with 200 ng/ml of inhibitor whereas its expression is decreased compared to the untreated snails. Intersection at x-axis, the y co-ordinate should be 1

Please indicate the prevalence for the snails treated and not treated with inhibitors. According to me, Figure 2C ca be added in supplementary data. Note that no legend is associated with the figure 2D

In the manuscript, the authors report that “Treatment of the snails with 464 lamivudine at day 14 post-exposure, however, showed some snails shed cercariae when 465 the drug was utilized later after infection.” Please indicate for how many snails (%) , this observation has been done. Also, if I understand well, this treatment can affect parasite development. So, in Fig5A, what is the developmental stage of the parasites in the BBO2 snails exposed to lamivudine ? Are they killed, encapsulated or present as sporosysts but unable to grow. Please provide further discussion /clarification.

MINOR COMMENTS

Reviewer 1

Suggested edits

Line 37-38 – swap permissive and non-permissive

Line 42 – join acetyl and transferase = acetyltransferase

Line 42 – need to separate histone acetyl transferase and histone deacetylase with comma

Line 139 – suggest changing “obtain leads” to “obtain candidates” to avoid having leads twice in same sentence

Line 194 – insert comma after “(TmHMM)

Line 203 – please include more detail on how specific transcripts to be examined vi two step qPCR were selected. 

Line 209 – not clear what “#7050” refers to. Should this citation be a number in refs rather than name and year?

Line 216 - join acetyl and transferase = acetyltransferase, insert comma after (BgHAT)

Line 228 – join “Bio” and “systems”. Is Wolston Warrington the correct address? I could not find it on a map. Maybe just include Warrington

Lines 208-226 – there is lack of consistency in wording of two step RT qPCR (i.e. sometimes two step included other times not), and whether given in full or abbreviated

Lines 244-75 – P of BgPiwi was uppercase previously

Line 246 – would change wording to “either dsRNA or siRNA, complexed with PEI”

Line 274 – remove “in”

Line 286 – have already given Sigma address in full (line 252)

Line 286 – insert comma after “MO)”

Line 310 – should the comma be replaced with a semicolon? 

Line 317 & 328 – perhaps change “presented” to added

Line 330-1 – would reword description of incubations at different temperatures, e.g., The Top Brite was set for 2 min at 37C, then 2 mins at 75C, and then 30 min at 37C

Line 352 – numbers can be in same box, don’t’ need separate box for each number

Line 366 – De-novo doesn’t need uppercase d or a hyphen

Line 368 – should be one sentence, no period after “temperatures”

Line 395 – “demonstrates” rather than “demonstrate”

Line 399, “he” should be “the”

Line 426 – insert comma after “nimbus”

Line 428 – after “PEI)” insert a comma and then “ the”

Line 456 – remove “either”

Line 509 – insert space after “insects”

Line 514-19 – insert some commas in lists of genes, overall this sentence doesn’t read well

Line 528 – insert commas in list of transcripts (e.g. after (HAT) and after (HDAC)

Line 546 – replace “preformed” with “examined” or “recorded”

Line 551 – delete “reveals the repertoire”

Line 592 – can put 2 numbers in same [ ]

Figure legends

Line 630 & 682 & 695 & 716 & 728 – think plural of asterisk is asterisks, not asterixis

Line 674 -5 – legend refers to blue and gray histograms but both are blue though different shades of blue

Figure 2D lacks a figure legend

Figure 6 legend could clarify A, B, and C as showing examples of shell positions. Clarify which part of legend referring to 6D.

Reviewer 2

L2 : 25°C (permissive) maybe non permissive ?

Please harmonize writing for PIWI, Piwi or piwi

Please indicate the origin of BS90 (BS-90 ?) as done for the other snail strains.

L193 : Pfam

L196 : clarify different categories

L188 : 2 hours ?

L209 : Format the reference 

L271 Edit a space between TR and nimbus

L376 and 396 : hours ?

Addressing these points may require minor revisions and supplementary experiments, but will significantly improve the quality of the manuscript, which has great potential to provide novel insight into Biomphalaria - Schistosoma complex interaction

PLOS authors have the option to publish the peer review history of their article (what does this mean?). If published, this will include your full peer review and any attached files.

Reviewer #1: No

Reviewer #2: No
---

## [Decision Letter · Decision Letter 1]

16 Jun 2021

Dear Dr. Knight,

Thank you very much for submitting your manuscript "PIWI silencing mechanism involving the retrotransposon nimbus orchestrates resistance to infection with Schistosoma mansoni in the snail vector, Biomphalaria glabrata" for consideration at PLOS Neglected Tropical Diseases. As with all papers reviewed by the journal, your manuscript was reviewed by members of the editorial board and by several independent reviewers. The reviewers appreciated the attention to an important topic. Based on the reviews, we are likely to accept this manuscript for publication, providing that you make the minor corrections according to the review recommendations. Please address the issue of qPCR.

Sincerely,

Philip T. LoVerde

Guest Editor

Michael Hsieh

Deputy Editor

Reviewer's Responses to Questions

**Key Review Criteria Required for Acceptance?**

**Methods**

-Are the objectives of the study clearly articulated with a clear testable hypothesis stated?

-Is the study design appropriate to address the stated objectives?

-Is the population clearly described and appropriate for the hypothesis being tested?

-Is the sample size sufficient to ensure adequate power to address the hypothesis being tested?

-Were correct statistical analysis used to support conclusions?

-Are there concerns about ethical or regulatory requirements being met?

Reviewer #1: (No Response)

Reviewer #2: all the points are satisfied

**Results**

-Does the analysis presented match the analysis plan?

-Are the results clearly and completely presented?

-Are the figures (Tables, Images) of sufficient quality for clarity?

Reviewer #1: (No Response)

Reviewer #2: y

**Conclusions**

-Are the conclusions supported by the data presented?

-Are the limitations of analysis clearly described?

-Do the authors discuss how these data can be helpful to advance our understanding of the topic under study?

-Is public health relevance addressed?

Reviewer #1: (No Response)

Reviewer #2: y

**Editorial and Data Presentation Modifications?**

Reviewer #1: line 44 - differential expression doesn't need uppercase letters (Differential Expression)

line 83 - "became" should be "become"

line 109 - check if unnecessary space between "long-" and "term"

line 155 - insert "temperatures" after (25C)

lines 166-168 - could these 2 sentences be merged, in 1st sentence state that BS-90 resistant to S. mansoni NMRI strain, and in 2nd sentence state that BS-90 resistant to S. mansoni strains from Old World and New World. 

line 177 - should "exposure" be plural?

line 180 - remove space in bri address

line 211 - transcriptome should be plural as multiple transcriptomes generated

line 214 - insert comma after "(32C)

line 223 - Table 1 lacks a title and not clear if upregulated and downregulated refers to in 32C or in 25C. Legend for table would be helpful.

line 228 - why is the formatting of this citation not a number?

line 235 - check if unnecessary space between "RNA-" and "seq"

line 237 - insert "temperatures" after "(25C)"

line 267 - give BLAST in full earlier (line 234)

line 289 - check for unnecessary spaces within equation in subscripts

line 311 - not clear which reagent "(Sigma)" refers to

line 314 - don't need "(" as have one line 313

line 318 - don't need uppercase T for "Total"; don't think it is necessary to include "(by serendipity)"

line 329 - please correct "in at"

line 349 - please check if need space in siRNABgPIWI/

line 383 - suggest changing verb from "was presented to a coverslip" to "was applied to a coverslip"

line 427 - unnecessary space after "mansoni"

line 450 - "No" should be "No."

line 451 - unnecessary space after "set"

line 454 and 455 - check if space missing between "1" and "and"

line 468 - title for table not accurate as table not showing multiple alignments, it shows the degree of identity between homologs

line 471 - suggest replacing "however" with "furthermore"

line 482 - suggest changing "infection" to exposure as BS-90 maintained at 25C

line 497 - insert commas after "transfected" and "normal snails". Also suggest not using the term "normal". Instead suggest using "untransfected". Same for line 509 regarding use of "normal".

line 518 and following - lack of consistency in BgPiwi, p often lowercase in remainder of manuscript (e.g., lines 525, 526, 533, 534 )

lines 553 & 554 - suggest using S. mansoni miracidia/sporocysts instead of "parasite" for sake of clarity and consistency 

line 612 - insert comma after "(BgHAT)"

line 679 - remove space after "54" and insert before "55"

line 684 - suggest changing wording to clarify that referring to exposure of resistant and infection of susceptible, rather than referring to both using "infection"

line 681 - check for consistency in how write piwi when referring to gene 

line 686 - suggest changing miracidium to plural

line 687 - do the authors mean systematic rather than systemic?

Reviewer #2: Accept

**Summary and General Comments**

Reviewer #1: This manuscript describes research that was done to increase our understanding of the complex molecular interactions between B. glabrata snails and Schistosoma mansoni. It is of interest to researchers of this host-parasite system, as well as to those in the fields of host-parasite interactions, molecular parasitology, and vector biology, to name a few. The research is very novel, interesting, and significant to the fields mentioned above. 

I have suggested multiple edits in a separate section as well as a few concerns or suggestions below. 

lines 247-250 - It seems that if qPCR carried out to verify RNA-seq data from BS-90 maintained at 25C and 32C, then qPCR should also be carried out on BS-90 maintained at 32C. It is valuable to carry out qPCR on susceptible BBO2 but i think it should also be carried out on BS-90 maintained at 32C seeing as original RNA-seq data from BS-90 maintained at 32C. Lines 420-430 mention carrying out qPCR on BS-90 maintained at 32C but earlier in manuscript, and in text following from lines 430 it is not clear when BS-90 maintained at 32C used for qPCR. 

Line 457 - BgPiwi described as being highly conserved by amino acid identity around 40% which isn't that high. Would just state that it is conserved.

Reviewer #2: Authors have answered point by point to the several questions that i have raised. 

The manuscript can be published

PLOS authors have the option to publish the peer review history of their article (what does this mean?). If published, this will include your full peer review and any attached files.

Reviewer #1: No

Reviewer #2: No

Figure Files:

Data Requirements:

Reproducibility:

References

---

## [Editor Report · Decision Letter 2]

27 Jul 2021

Dear Dr. Knight,

We are pleased to inform you that your manuscript 'PIWI silencing mechanism involving the retrotransposon nimbus orchestrates resistance to infection with Schistosoma mansoni in the snail vector, Biomphalaria glabrata' has been provisionally accepted for publication in PLOS Neglected Tropical Diseases.

Best regards,

Philip T. LoVerde

Guest Editor

Michael Hsieh

Deputy Editor

---

## [Editor Report · Acceptance letter]

24 Aug 2021

Dear Dr. Knight,

We are delighted to inform you that your manuscript, "PIWI silencing mechanism involving the retrotransposon nimbus orchestrates resistance to infection with *Schistosoma mansoni* in the snail vector, *Biomphalaria glabrata*," has been formally accepted for publication in PLOS Neglected Tropical Diseases.

Best regards,

Shaden Kamhawi

co-Editor-in-Chief

Paul Brindley

co-Editor-in-Chief
